# PISA: Privacy-Preserving Split Adaptation with Model IP Protection

**Haocheng Yang** [1] **Xiang Cheng** [1] **Zongda Han** [2] **Pengjie Wang** [1] **Changkang Chi** [1] **Pengfei Zhang** [3 4] **Sen Su** [1]

## Abstract

Fine-tuning Large Language Models (LLMs) enables data holders to construct proprietary, task-specific models by leveraging external high-performance computing infrastructure. However, existing paradigms typically address data privacy and model intellectual property (IP) in isolation, failing to simultaneously uphold both constraints. Privacy-prioritized methods compromise model IP by hosting parameters remotely, while IP-oriented collaborative schemes relying on end-to-end gradient flows inherently violate strict data privacy standards. To address these challenges, we present **PISA** (**P**rivacy-preserving and **IP**-protected **S**plit **A**daptation), a split fine-tuning framework designed to preserve both data privacy and model IP while maintaining high utility. In PISA, we propose three methods: a Manifold Rectification Pre-training (MRP) method to equip the server-side model with intrinsic robustness against privacy-induced distribution shifts; a Dual-Stream Semantic Compensation (DSC) method to recover feature utility using local clean data as priors; and a Utility-Aware Gradient Rectification (UGR) method to adaptively maximize the performance of the parameter-constrained local model. Experiments on GLUE show that PISA ensures dual protection and delivers a substantial 23.0% performance gain over the privacy-prioritized baseline under strict privacy budgets.

## 1. Introduction

In sensitive domains such as healthcare and finance, leveraging the generalization capabilities of Large Language Models (LLMs) for downstream task fine-tuning has become an increasingly widespread practice (Wu et al., 2023; Shang et al., 2026; Yang et al., 2026). However, due to the inherent asymmetry in computational resources and data assets, a collaborative paradigm has emerged: a server (e.g., a cloud service provider) possessing massive computing infrastructure and pre-trained foundation models provides fine-tuning support to a data holder (e.g., a healthcare institution) possessing private domain data and aiming to establish task-specific model intellectual property (IP). Nevertheless, this collaboration is bound by dual stringent constraints. On one hand, data privacy prohibits the transmission of raw sensitive data outside the trusted domain. On the other hand, the task-specific model IP, specifically the fine-tuned task-specific parameters representing the core business logic, must strictly reside locally to prevent misappropriation by the service provider (Zhao et al., 2025).

Existing paradigms fail to reconcile these two objectives, typically addressing one at the expense of the other. Privacy-prioritized methods (Yue et al., 2021; Chen et al., 2023a; Du et al., 2023), anchored in the gold standard of Local Differential Privacy (LDP) (Duchi et al., 2013), typically outsource model training to the server via input sanitization, thereby exposing the fine-tuned model IP. Conversely, IP-oriented schemes (Vepakomma et al., 2018; Wang et al., 2025) retain local control of the model but rely on end-to-end gradient back-propagation, which inherently leaks private data to the server. Consequently, there is currently no unified framework that can simultaneously satisfy both constraints within a cohesive adaptation paradigm.

To address these dual security requirements while leveraging external compute, a possible solution is to adopt the reverse split architecture. In this architecture, the computation-intensive backbone layers (referred to as the Head) are retained on the server to provide feature extraction services, while the private task-specific sub-model (referred to as the Tail, comprising the final backbone layers and the classifier) is strictly deployed on the data holder side. Crucially, to prevent IP leakage via gradient back-propagation, it enforces a gradient blockage protocol, where the data holder withholds

---

[1]State Key Laboratory of Networking and Switching Technology, Beijing University of Posts and Telecommunications, Beijing, China [2]China Mobile Group Hunan Company Ltd., Changsha, Hunan, China [3]Key Laboratory of Computing Power Network and Information Security, Ministry of Education, Qilu University of Technology (Shandong Academy of Sciences) [4]School of Computer Science and Engineering, Anhui University of Science and Technology, Huainan, 232001, China. Correspondence to: Xiang Cheng <chengxiang@bupt.edu.cn>.

*Proceedings of the 43$^{rd}$ International Conference on Machine Learning*, Seoul, South Korea. PMLR 306, 2026. Copyright 2026 by the author(s).

backward gradients. However, this unidirectional information flow compels the server Head to remain frozen, as it is deprived of the optimization signals required for parameter updates. We identify that this structurally constrained architecture suffers from a critical "Frozen-Head, Limited-Tail" bottleneck: the frozen server model cannot adapt to the severe distribution shift caused by the LDP noise (Ye et al., 2023), and the lightweight local Tail (typically < 20% capacity) lacks the representational capacity to rectify the massive semantic distortion transmitted by the Head (Wang et al., 2017). Consequently, such implementation inevitably leads to model convergence failure and utility collapse.

To address this challenge, we propose **PISA** (**P**rivacy and **IP S**plit **A**daptation), a framework that implements prior-guided semantic restoration by capitalizing on the data holder's exclusive access to local clean raw data. PISA systematically resolves the tension through three synergistic methods. First, to overcome the adaptability limitation of the frozen server, the Manifold Rectification Pre-training (MRP) method enforces an intrinsic robustness within the server Head. By employing a multi-task manifold alignment objective, MRP enables the server Head to actively remap the LDP-distorted embeddings back to the clean natural language manifold prior to collaboration. Second, to counteract the severe semantic shattering caused by input perturbation, the Dual-Stream Semantic Compensation (DSC) method restores feature utility. It leverages the data holder's unique access to clean raw data as a "semantic anchor" to construct a dynamic residual patch, effectively steering the shattered feature representations back towards their intended semantic direction without consuming additional privacy budget. Finally, to stabilize the optimization of the parameter-constrained Tail, the Utility-Aware Gradient Rectification (UGR) method mitigates the impact of gradient toxicity. It utilizes the server's linguistic priors to construct an unsupervised quality gate that analytically adapts the backward gradient flow, ensuring that the limited parameter updates are driven solely by high-confidence supervision signals.

The main contributions of this work are as follows:

- We present PISA, a split adaptation framework that pioneers a solution to simultaneously uphold strict data privacy and structural model IP protection while ensuring high utility.

- We propose a Manifold Rectification Pre-training (MRP) method that effectively mitigates privacy-induced distribution shifts while maintaining the intrinsic robustness of the server-side representation.

- We put forward a Dual-Stream Semantic Compensation (DSC) method that effectively corrects for feature

distortion to recover the semantic information lost due to noise perturbation.

- We design a Utility-Aware Gradient Rectification (UGR) method that effectively prevents optimization divergence in the parameter-constrained Tail while maximizing the adaptation efficiency.

- Extensive experiments on representative GLUE benchmark datasets demonstrate that PISA effectively bridges the gap between privacy and IP protection, delivering utility comparable to single-objective baselines while adhering to stricter constraints.

## 2. Related Work

### 2.1. Privacy-Preserving Fine-Tuning Paradigms

With the widespread deployment of Large Language Models (LLMs) in sensitive domains, privacy leakage during the adaptation phase has emerged as a critical concern. Existing literature predominantly prioritizes strict data privacy compliance, often at the expense of model IP. A major stream of research employs text sanitization, utilizing LDP mechanisms to desensitize text prior to transmission. For instance, heuristic approaches attempt to anonymize specific entities (Kan et al., 2023; Chen et al., 2023b), while more rigorous methods (Yue et al., 2021; Awon et al., 2025) adopt LDP to map tokens into a perturbed semantic space. While effective for data protection, these methods inherently necessitate a server-hosted training paradigm, where the sanitized data is offloaded to the server for model updating. Similarly, feature-perturbation approaches (Du et al., 2023; Li et al., 2025; Shen et al., 2025) perturb intermediate embeddings to mask raw attributes but still rely on the server to execute the bulk of the training workload. This structural dependency compels data holders to surrender control over fine-tuned parameters to the server to satisfy privacy or computational constraints. Since the server hosts the resulting model encapsulating proprietary task logic, this paradigm inherently forfeits model IP.

### 2.2. Split Learning and Model IP Protection

To reclaim model IP and enable local adaptation, Split Learning (SL) has been introduced as a collaborative framework that physically decouples the model into client-side and server-side segments (Gupta & Raskar, 2018; Vepakomma et al., 2018). By retaining a portion of the network within the local trusted domain, SL conceptually allows the data holder to assert ownership of the task-specific sub-model. While recent works (Wang et al., 2025) adapt SL for LLMs to reduce communication costs, they remain structurally reliant on gradient exchange. However, relying on end-to-end gradient flow introduces critical vulnerabilities as recent security audits (Liu et al., 2024) demonstrate that cross-

domain interaction creates a bidirectional leakage channel. Specifically, adversarial servers can exploit backward gradients to reconstruct private data (Pasquini et al., 2021) or effectively infer proprietary logic via model stealing attacks (Zhang et al., 2025), thereby undermining the assumption that split model alone ensures model IP confidentiality.

## 2.3. Cryptographic and Hardware-based Solutions

Beyond algorithmic perturbations, cryptographic protocols like Homomorphic Encryption (HE) and Multi-Party Computation (MPC) theoretically offer rigorous privacy guarantees without revealing raw data (Gilad-Bachrach et al., 2016). However, their prohibitive computational overhead and communication latency render them impractical for fine-tuning LLMs (Huang et al., 2022).

Moreover, Trusted Execution Environments (TEEs) attempt to isolate computation in hardware-based secure enclaves. However, TEEs are constrained by limited secure memory capacity relative to LLM sizes and remain vulnerable to advanced side-channel attacks (Yuan et al., 2024), failing to provide a scalable solution for efficient LLM adaptation.

## 3. Preliminaries

### 3.1. Local Differential Privacy

Local Differential Privacy (LDP) serves as the golden standard for privacy-preserving data collection in untrusted environments by ensuring local perturbation before transmission. Formally, $\epsilon$-LDP is defined as:

**Definition 3.1** ($\epsilon$-LDP). A randomized mechanism $\mathcal{M}$ satisfies $\epsilon$-Local Differential Privacy if for any pair of input values $x, x' \in \mathcal{X}$ and any output $y \in \text{Range}(\mathcal{M})$, the following inequality holds:

$$\Pr[\mathcal{M}(x) = y] \le e^\epsilon \cdot \Pr[\mathcal{M}(x') = y], \qquad (1)$$

where $\epsilon > 0$ denotes the privacy budget.

However, standard LDP mechanisms designed for numerical data prove inapplicable to textual data residing in high-dimensional semantic spaces. Therefore, we adopt $d_\chi$-privacy (Chatzikokolakis et al., 2013) as the formalism to provide rigorous privacy guarantees calibrated to the underlying geometry of the data space (Feyisetan et al., 2020).

**Definition 3.2** ($d_\chi$-privacy). Let $\mathcal{X}$ be the input domain and $\mathcal{Y}$ be the output domain equipped with a distance metric $d_\chi$ over $\mathcal{X}$. A stochastic mechanism $\mathcal{M} : \mathcal{X} \to \mathcal{Y}$ satisfies $\epsilon d_\chi$-privacy if for any pair of inputs $x, x' \in \mathcal{X}$, the probability distributions of outputs satisfy:

$$\Pr[\mathcal{M}(x) = y] \le e^{\epsilon \cdot d_\chi(x,x')} \cdot \Pr[\mathcal{M}(x') = y], \quad (2)$$

for all $y \in \mathcal{Y}$, where $\epsilon > 0$ represents the privacy budget.

## 3.2. Problem Statement

**System Model.** We consider an asymmetric collaboration framework involving a resource-constrained data holder $\mathcal{C}$ and a resource-abundant server $\mathcal{S}$. To simultaneously reconcile data privacy regulations with model IP protection, we establish a reverse split architecture equipped with a strict gradient blockage. In this architecture, data privacy is enforced by sanitizing raw inputs via the LDP mechanisms before any transmission occurs. Conversely, model IP protection is structurally guaranteed by a dual safeguard that involves physically localizing task-specific parameters within the data holder's trusted domain while enforcing a gradient blockage protocol to prevent the server from inferring proprietary logic via backward optimization signals.

Specifically, the data holder possesses a private dataset $\mathcal{D}_\mathcal{C} = \{(x_i, y_i)\}_{i=1}^N$. Notably, the task labels $y_i$ are strictly confined within the local trusted domain throughout the entire lifecycle. For the raw input $x_i$, an LDP mechanism $\mathcal{M}$ is applied to generate a perturbed representation $\tilde{x}_i = \mathcal{M}(x_i)$ prior to transmission. To facilitate feature extraction, the system utilizes a large-scale pre-trained foundation model split into two sub-models: The computation-intensive backbone, denoted as the Head $F_\phi$ (parameterized by $\phi$), is retained on the server; The task-specific layers, denoted as the Tail $G_\theta$ (parameterized by $\theta$), are distributed to the data holder to serve as the private sub-model. The collaborative data flow is formulated as $\hat{y} = G_\theta(F_\phi(\tilde{x}))$. During the forward pass, the server extracts features $h = F_\phi(\tilde{x})$ and transmits them to the data holder. During training, the data holder computes the task loss $\mathcal{L}(\hat{y}, y)$ and updates $\theta$ locally, but withholds any backward gradients from the server. Consequently, the server parameters $\phi$ remain frozen, ensuring that no derivative information regarding the task label or model parameters flows back to the server.

**Threat Model.** We assume an honest-but-curious server that adheres to the collaboration protocol but attempts to infer private data content or steal the proprietary task model $G_\theta$. Under our protocol, the server's view is strictly limited to the sanitized input and its own model parameters, denoted as $\mathcal{V}_\mathcal{S} = \{\tilde{x}, F_\phi\}$. The adversary aims to either reconstruct the raw text $x$ from $\tilde{x}$ (i.e., a Data Privacy attack) or infer the parameters and functionality of $G_\theta$ (i.e., a Model IP attack). Our goal is to maximize the utility of the fine-tuned Tail $G_\theta$ while satisfying strict LDP guarantees for data and structurally preventing model stealing by eliminating the backward leakage channel.

## 4. Methodology

### 4.1. Framework Overview

As illustrated in Figure 1, the execution lifecycle of PISA is organized into two decoupled phases. To mitigate the

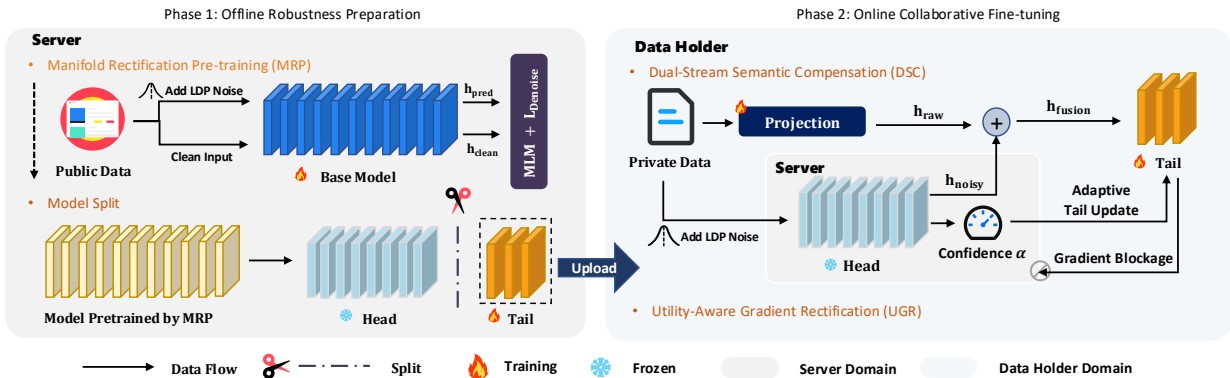

*Figure 1.* Overview of the PISA framework architecture.

utility degradation caused by the reverse split architecture, we integrate three specialized technical methods into the workflow to systematically restore model utility.

In the remainder of this section, we provide a detailed elaboration of the specific procedures and technical designs implemented within these two phases.

**Phase I: Offline Robustness Preparation.** Prior to collaboration, the server resolves the distribution mismatch between standard pre-trained features and privacy-induced noise. The server independently executes the Manifold Rectification Pre-training (MRP) method (Section 4.2) using a public dataset $\mathcal{D}_{pub}$ to equip the full foundation model with intrinsic robustness against LDP perturbations. Upon convergence, the model undergoes a physical decoupling protocol, where the robust backbone is retained on the server as the frozen Head $F_\phi$, while the task-specific layers are detached to instantiate the Tail $G_\theta$, which is then distributed to the data holder.

**Phase II: Online Collaborative Fine-tuning.** During the fine-tuning phase, the system executes a strictly asymmetric interaction where the server provides feature extraction services for the LDP-perturbed inputs $\tilde{x}_i$, while the data holder updates its private Tail locally. To address the information bottleneck in this process, we introduce two synergistic calibration methods. First, targeting semantic shattering in forward propagation, the Dual-Stream Semantic Compensation (DSC) method (Section 4.3) implements feature-level calibration by leveraging local clean priors to repair global noisy representations. Second, addressing gradient toxicity in backward propagation, the Utility-Aware Gradient Rectification (UGR) method (Section 4.4) implements optimization-level calibration by reusing server-side confidence estimates to rescale local updates.

Furthermore, we also provide a privacy analysis (detailed in Appendix A) and a theoretical complexity analysis (detailed in Appendix B) of PISA.

## 4.2. Manifold Rectification Pre-training (MRP)

In this section, we elaborate on the construction of the Manifold Rectification Pre-training (MRP) method. In the collaborative fine-tuning framework, LDP mechanisms achieve privacy preserving through discrete text replacement. However, this process inevitably induces severe distribution shifts and semantic shattering. Grounded in the manifold hypothesis (Bengio et al., 2013; Aghajanyan et al., 2021), which posits that valid natural language representations are concentrated on low-dimensional semantic manifolds, we identify that standard pre-trained models, optimized for clean text, suffer from performance collapse under privacy-induced shifts. This occurs because perturbed inputs act as "off-manifold" outliers that deviate significantly from the intrinsic latent structure. To address this, we propose to pre-construct a feature encoder equipped with intrinsic robustness, capable of implicitly projecting these noisy outliers back onto the valid semantic manifold.

Specifically, to strictly adhere to the privacy protocol, this phase is conducted entirely on the server using a public auxiliary dataset $\mathcal{D}_{pub}$ (e.g., Wikipedia), without accessing the data holder's private data $\mathcal{D}_C$. Let $\mathcal{M}$ denote the full foundation model instance deployed on the server for pre-training. We designate a specific cut-layer depth $K$ to define the future split point. Consequently, $\mathcal{M}$ can be conceptually viewed as a composition $\mathcal{M}(\cdot) = G_{\theta_{init}}(F_\phi(\cdot))$, where $F_\phi$ represents the backbone layers (layers 1 to $K$) and $G_{\theta_{init}}$ represents the sub-model (layers $K + 1$ to the final output). We define a multi-task adversarial learning objective that synergistically optimizes Masked Language Modeling (MLM) and semantic alignment. Given a public sample $x \in \mathcal{D}_{pub}$ and its perturbed version $\tilde{x} = \mathcal{M}_{LDP}(x)$, we define the robust representation $h_{pred} = F_\phi(\tilde{x})$ extracted by the backbone and the clean reference $h_{clean} = F_\phi(x)$. The joint loss function $\mathcal{L}_{Phase\_I}$ is formulated as:

$$\mathcal{L}_{Phase\_I} = \mathcal{L}_{MLM}(\tilde{x}) \\ + \lambda\mathcal{L}_{Denoise}(h_{pred}, h_{clean}) \quad (3)$$

where $\mathcal{L}_{Denoise}$ adopts a hybrid metric composed of structured euclidean distance and cosine directional constraints. This objective serves as a geometric rectification signal, forcing the distorted features $h_{pred}$ to realign with the "on-manifold" clean priors $h_{clean}$ in terms of both spatial position and semantic direction:

$$
\begin{aligned}
\mathcal{L}_{Denoise} = {} & \lambda_{l2}\|h_{pred} - h_{clean}\|_2^2 \\
& + \lambda_{cos}\big(1 - \cos(h_{pred}, h_{clean})\big)
\end{aligned}
\tag{4}
$$

Upon convergence, we enforce the reverse split topology via physical partitioning. The optimized full model $\mathcal{M}$ is split at the designated depth $K$ into two disjoint modules: the bottom backbone is frozen as the service Head $F_\phi$ to provide stable denoising features, while the upper transformer layers and the task head (originally $G_{\theta_{init}}$) are detached to instantiate the Tail $G_\theta$. This initialized Tail $G_\theta$ is then transmitted to the data holder, ensuring that the client-side model starts with a representation space aligned with the server's robust features.

### 4.3. Dual-Stream Semantic Compensation (DSC)

Constrained by the gradient truncation of the asymmetric protocol, the server is forced to freeze the Head, which accounts for the vast majority of the model's parameters. While this design facilitates decoupling, it introduces a severe adaptation bottleneck: relying solely on fine-tuning the local Tail module—which typically contains minimal parameters—results in insufficient model capacity to fully digest the massive and irreversible information loss caused by the differential privacy mechanism. In this capacity-constrained regime, attempting to infer or compensate for lost original semantic information relying purely on the tail module is mathematically intractable, leading to a severe performance ceiling during the collaborative fine-tuning phase. Therefore, we introduce the Dual-Stream Semantic Compensation (DSC) method utilizing the clean raw data held locally by data holder as a semantic anchor for real-time calibration.

Specifically, the data holder first constructs a lightweight projection layer $\mathcal{P}$ to extract the residual representation $h_{raw}$. To obtain the input features, the data holder maintains a local replica of the token embedding layer to generate $e_{raw}$ directly from the private text $x$. This layer takes the local raw embeddings $e_{raw}$ (corresponding to $x$) as input and performs a non-linear mapping defined as:

$$
\begin{aligned}
h_{raw} &= \mathcal{P}(e_{raw}; \Theta_p) \\
&= W_2 \cdot \sigma_{GELU}\big(\text{LN}(W_1 e_{raw} + b_1)\big) + b_2
\end{aligned}
\tag{5}
$$

where LN denotes Layer Normalization and $\sigma_{GELU}$ represents the non-linear activation function, utilized to enhance the non-linear expressive capability of features and

achieve semantic alignment with the latent space of the Head. Subsequently, we treat the server-side perturbed features $h_{noisy} = F_\phi(\tilde{x})$ and the local clean representation $h_{raw}$ as heterogeneous information streams, implementing semantic rectification via a fusion operator $\Phi$:

$$
h_{fusion} = \Phi(h_{noisy}, h_{raw})
\tag{6}
$$

where $h_{raw}$ is defined as a residual patch that augments model representation. This design combining global guidance with local anchoring, allows the model to dynamically calibrate privacy deviations using local prior knowledge while preserving global context. By confining the raw input and its derived representations strictly within the local environment without involving gradient back-propagation, this method effectively mitigates the capacity shortage caused by asset protection mechanisms while ensuring that no additional sensitive information is exposed to the server during the collaborative process.

### 4.4. Utility-Aware Gradient Rectification (UGR)

In the parameter-constrained local Tail fine-tuning phase, random perturbations introduced by the differential privacy mechanism not only cause semantic deviations during forward propagation but also induce significant gradient toxicity during back-propagation. Due to the discrete replacement characteristic, samples that are severely perturbed generate outlier gradients that deviate from the true optimization direction. For the Tail with extremely limited parameters, these high-variance toxic gradients can easily dominate the weight update direction, causing the model to oscillate or even diverge on the optimization manifold. Therefore, we propose the Utility-Aware Gradient Rectification (UGR) method, which implements dynamic regulation of sample contribution through a confidence-guided quality gating mechanism.

Specifically, the server reuses the MLM head $\mathcal{H}_{MLM}$ pretrained in the first phase as a discriminator, leveraging its capability to model the natural language manifold to evaluate the semantic integrity of the noisy features $h_{noisy}$. We define the quality confidence $\alpha$ of a sample as its maximum posterior probability over the vocabulary distribution:

$$
\alpha = \max_{w \in \mathcal{V}} \text{Softmax}(\mathcal{H}_{MLM}(h_{noisy}))_w
\tag{7}
$$

The server then transmits this scalar $\alpha$ alongside the feature $h_{noisy}$ to the data holder. Based on this quality assessment, the UGR utilizes $\alpha$ as a scalar coefficient to perform real-time rescaling of the back-propagated gradients. The parameter update rule is formalized as:

$$
\theta_{Tail} \leftarrow \theta_{Tail} - \eta \cdot \alpha \cdot \nabla_\theta \mathcal{L}_{task}(h_{fusion}, y)
\tag{8}
$$

where $\theta_{Tail}$ represents the learnable parameters of the local Tail module, and $\eta$ denotes the learning rate. Mathemati-

cally, UGR constructs a soft denoising filter, where high-confidence samples are assigned full weight to guide the optimization direction, while the gradient contribution of low-confidence samples is significantly suppressed. This ensures that the model can extract robust feature representations from a limited number of effective samples even in scenarios with extremely low privacy budgets.

To provide theoretical insights into the noise mitigation mechanism of UGR, we analyze the convergence properties of the optimization process under simplified assumptions (detailed proof in Appendix C).

**Assumption 4.1** (Regularity Conditions). *To theoretically analyze the convergence of our framework, we follow standard stochastic optimization literature* (Bottou et al., 2018) *and adopt the following assumptions regarding the task loss function $\mathcal{L}_{task}$ and the stochastic gradient estimator:*

1. *$L$-**Smoothness:** The expected loss function $f(\theta) := \mathbb{E}[\mathcal{L}_{task}(\theta)]$ is differentiable and $L$-smooth, i.e., $\|\nabla f(\theta_1) - \nabla f(\theta_2)\| \le L\|\theta_1 - \theta_2\|$ for all $\theta_1, \theta_2 \in \mathbb{R}^d$.*

2. **Unbiasedness:** *The stochastic gradient $g(\theta)$ is an unbiased estimator of the full gradient, satisfying $\mathbb{E}[g(\theta)] = \nabla f(\theta)$.*

3. **Bounded Variance:** *The variance of the stochastic gradients, which encapsulates the distortion from differential privacy noise, is bounded by a constant $\sigma_{noise}^2$, such that $\mathbb{E}[\|g(\theta) - \nabla f(\theta)\|^2] \le \sigma_{noise}^2$.*

**Theorem 4.2** (Weighted Convergence of UGR). *Consider the UGR update rule $\theta_{t+1} = \theta_t - \eta \tilde{g}_t$, where $\tilde{g}_t = \alpha_t \cdot g(\theta_t)$ is the rectified gradient with quality score $\alpha_t \in [0, 1]$. Let the average effective quality $\bar{\alpha}$ be defined as:*

$$\bar{\alpha} = \frac{1}{T}\sum_{t=1}^{T}\mathbb{E}[\alpha_t] > 0. \tag{9}$$

*Under Assumption 4.1, with a decaying step size $\eta = \frac{1}{\sqrt{T}}$, the weighted average gradient norm, defined as:*

$$\mathcal{G}_T^2 \triangleq \frac{\sum_{t=1}^{T}\mathbb{E}[\alpha_t\|\nabla f(\theta_t)\|^2]}{\sum_{t=1}^{T}\mathbb{E}[\alpha_t]}, \tag{10}$$

*satisfies the following convergence bound:*

$$\mathcal{G}_T^2 \le \frac{C_0}{\sqrt{T}} + \frac{C_1}{\sqrt{T}}\left(\frac{1}{T}\sum_{t=1}^{T}\frac{\mathbb{E}[\alpha_t^2]}{\bar{\alpha}}\sigma_{noise}^2\right) \tag{11}$$

*where $C_0$ and $C_1$ are positive constants depending on the initial loss gap and the smoothness constant $L$.*

*This bound explicitly demonstrates that the convergence rate is $\mathcal{O}(1/\sqrt{T})$. Notably, the noise variance term is scaled by*

*the squared quality score $\alpha_t^2$, ensuring that toxic samples (where $\alpha_t \to 0$) contribute negligibly to the optimization error, thereby providing a theoretical basis for the stability observed under strict privacy constraints.*

**Extension to Decoder-Only Architectures.** While PISA is instantiated with a BERT-base encoder, the framework generalizes naturally to decoder-only LMs. For MRP, $\mathcal{L}_{MLM}$ is replaced by a causal language modeling objective $\mathcal{L}_{CLM}$, while $\mathcal{L}_{Denoise}$ remains unchanged. For UGR, the confidence score $\alpha$ in Eq. 7 is computed as the sequence-level average log-likelihood over the Head output, generalizing the token-level max posterior to causal architectures.

## 5. Experiments

### 5.1. Experimental Setup

**Datasets.** Following established evaluation protocols in prior studies (Li et al., 2025; Du et al., 2023), we conduct experiments on three representative benchmarks from the GLUE (Wang et al., 2018): SST-2 for single-sentence sentiment analysis, QNLI for question-answering inference, and MNLI for multi-genre natural language inference. This selection covers diverse linguistic tasks ranging from simple classification to complex semantic matching. Detailed dataset statistics are provided in Appendix D.

**Baselines.** We benchmark PISA against three categories of methods to comprehensively evaluate the trade-offs among utility, privacy, and model IP. First, to establish theoretical performance bounds, we employ Vanilla FT which performs standard fine-tuning in a constraint-free setting. Second, for comparisons with existing privacy-prioritized paradigms, we select Hosted-LDP which adopts the general metric local differential privacy perturbation mechanism utilized in state-of-the-art works (Awon et al., 2025; Yue et al., 2021). Third, to assess structural IP protection without privacy noise, we utilize Naive-RS as a straightforward baseline which implements the proposed reverse split architecture with strict gradient blockage. Finally, to quantify the contribution of specific technical modules, we conduct an ablation study on two variants of our framework denoted as PISA- and PISA-*. Specifically, PISA- retains only the MRP method to evaluate the impact of offline robustness preparation, while PISA-* simultaneously integrates MRP and DSC method.

**Implementation Details.** Following existing works in privacy-preserving adaptation (Li et al., 2025; Mai et al., 2024), we employ bert-base-uncased (Devlin et al., 2019) and Llama2-7B (Touvron et al., 2023) as the foundation backbone, simulated on NVIDIA A100 GPUs. The system follows a strict reverse split architecture where the server-side head is pre-trained on the Wikipedia corpus (Reese et al., 2010). By default, we configure the split point at the 9th layer, designating the bottom 9 layers as the server Head

*Table 1.* Main results on GLUE benchmarks under varying privacy budgets ($\varepsilon$), reported as mean$_{std}$ over 5 runs.

| Method | SST-2 | | | QNLI | | | MNLI | | | Average | | |
|---|---|---|---|---|---|---|---|---|---|---|---|---|
| | $\varepsilon=10$ | $\varepsilon=12$ | $\varepsilon=14$ | $\varepsilon=10$ | $\varepsilon=12$ | $\varepsilon=14$ | $\varepsilon=10$ | $\varepsilon=12$ | $\varepsilon=14$ | $\varepsilon=10$ | $\varepsilon=12$ | $\varepsilon=14$ |
| *Theoretical Upper Bound* | | | | | | | | | | | | |
| Vanilla FT (Non-DP) | $93.0_{0.2}$ | $93.0_{0.2}$ | $93.0_{0.2}$ | $91.5_{0.3}$ | $91.5_{0.3}$ | $91.5_{0.3}$ | $84.5_{0.4}$ | $84.5_{0.4}$ | $84.5_{0.4}$ | $89.6_{0.3}$ | $89.6_{0.3}$ | $89.6_{0.3}$ |
| *Privacy-Prioritized* | | | | | | | | | | | | |
| Hosted-LDP | $54.5_{4.5}$ | $59.4_{2.8}$ | $86.4_{1.5}$ | $53.5_{2.1}$ | $56.8_{1.8}$ | $84.2_{1.2}$ | $37.0_{1.9}$ | $51.1_{1.4}$ | $81.0_{0.9}$ | $48.3_{2.8}$ | $55.7_{2.0}$ | $83.8_{1.2}$ |
| *IP-Prioritized* | | | | | | | | | | | | |
| Naive-RS | $51.6_{0.2}$ | $51.6_{0.1}$ | $51.6_{0.1}$ | $50.9_{0.2}$ | $50.9_{0.1}$ | $50.9_{0.1}$ | $33.6_{0.3}$ | $33.6_{0.2}$ | $33.6_{0.1}$ | $45.4_{0.2}$ | $45.4_{0.1}$ | $45.4_{0.1}$ |
| *Ours* | | | | | | | | | | | | |
| PISA- | $70.5_{1.5}$ | $76.7_{1.2}$ | $77.8_{0.9}$ | $53.9_{1.8}$ | $60.3_{1.4}$ | $61.6_{1.1}$ | $61.4_{1.4}$ | $62.3_{1.2}$ | $66.9_{1.0}$ | $61.9_{1.6}$ | $66.4_{1.3}$ | $68.7_{1.0}$ |
| PISA-* | $80.4_{1.1}$ | $82.5_{0.9}$ | $85.3_{0.6}$ | $60.9_{1.3}$ | $61.0_{1.1}$ | $71.6_{0.8}$ | $62.6_{1.2}$ | $62.8_{1.0}$ | $70.2_{0.7}$ | $67.9_{1.2}$ | $68.7_{1.0}$ | $75.7_{0.7}$ |
| **PISA** | $\mathbf{81.2_{0.9}}$ | $\mathbf{84.5_{0.7}}$ | $\mathbf{87.1_{0.5}}$ | $\mathbf{64.3_{1.1}}$ | $\mathbf{68.2_{0.9}}$ | $\mathbf{73.2_{0.6}}$ | $\mathbf{68.6_{1.0}}$ | $\mathbf{69.8_{0.8}}$ | $\mathbf{74.2_{0.6}}$ | $\mathbf{71.3_{1.0}}$ | $\mathbf{74.1_{0.8}}$ | $\mathbf{78.1_{0.6}}$ |

and the top 3 layers as the local Tail. During the fine-tuning phase, we utilize the SanText mechanism (Yue et al., 2021) for privacy preservation. Comprehensive hyperparameters and architectural details are provided in Appendix E.

### 5.2. Main Results

We illustrate the evaluation results on three GLUE benchmarks in Table 1. The empirical evidence demonstrates that PISA effectively mitigates the utility degradation caused by the dual constraints of differential privacy and the reverse split architecture, particularly in strict privacy regimes.

**Performance Comparison with Baselines.** As indicated in Table 1, PISA exhibits significant superiority over the privacy-prioritized baseline, Hosted-LDP, especially under low privacy budgets. When $\varepsilon = 10$, PISA achieves an average accuracy of 71.3%, surpassing Hosted-LDP (48.3%) by a substantial margin of 23.0%. A similar trend is observed at $\varepsilon = 12$, where our method maintains a performance lead of 18.4%. This performance gap can be attributed to the structural advantage of PISA; while Hosted-LDP applies noise directly to embeddings without structural compensation, our framework actively repairs the semantic information destroyed by LDP noise through the proposed calibration modules. Furthermore, the comparison with the IP-prioritized baseline, Naive-RS reveals the inherent difficulty of the proposed asymmetric architecture. The Naive-RS method fails to converge effectively, yielding performance metrics close to random guessing (e.g., 51.6% on SST-2 and 33.6% on MNLI). This observation confirms that the strict gradient blockage creates an insurmountable information bottleneck for standard fine-tuning, validating the necessity of our specialized adaptation mechanisms to bridge the disconnected forward and backward passes.

Notably, we also provide a detailed efficiency comparison against standard baselines in Appendix F. Furthermore, we empirically validate PISA to decoder-only architectures on

Llama2-7B in Appendix I, where PISA consistently outperforms Hosted-LDP under strict privacy budgets.

**Privacy-Utility Trade-off.** To further investigate the dynamic relationship between privacy budgets and model utility, we visualize the performance trends of Hosted-LDP and PISA on SST-2 and QNLI datasets across a spectrum of $\varepsilon$ values in Figure 2. We observe a distinct crossover pattern in the performance curves at approximately $\varepsilon = 14$. In the strict privacy regime where $\varepsilon < 14$, PISA consistently outperforms Hosted-LDP, demonstrating that our method is highly robust against severe noise perturbations. Conversely, when $\varepsilon > 14$, Hosted-LDP exhibits a sharp performance recovery, beginning to match or surpass PISA. This phenomenon occurs because the frozen Head architecture in PISA, which is mandatory for IP protection, introduces a fixed capacity bottleneck that becomes the limiting factor when privacy noise is low. In contrast, Hosted-LDP benefits from full-parameter fine-tuning in relaxed privacy settings, allowing it to achieve higher utility limits, albeit at the cost of exposing the model IP. This trade-off underscores the unique position of PISA as an optimal solution for scenarios demanding both privacy and IP protection.

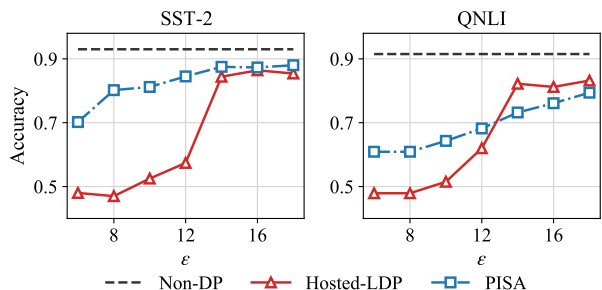

*Figure 2.* Main Results.

## 5.3. Ablation Study

The ablation results in Table 1 quantitatively validate the contribution of each technical module. The variant PISA$^-$, which retains only MRP, already achieves substantial improvements over Naive-RS across all settings. At $\varepsilon = 10$, PISA$^-$ attains an average accuracy of 61.9%, a gain of 16.5 points over Naive-RS (45.4%), confirming that equipping the server Head with intrinsic robustness against LDP-induced distribution shifts provides a meaningful initialization for the fine-tuning phase. However, the considerable gap relative to the full framework indicates that offline robustness preparation alone is insufficient to overcome the capacity bottleneck imposed by the lightweight Tail.

The introduction of DSC in PISA$^*$ yields a remarkable performance boost. At $\varepsilon = 10$, PISA$^*$ improves average accuracy from 61.9% to 67.9%, with the gain on SST-2 reaching 9.9 points (80.4% vs. 70.5%), demonstrating that real-time feature-level calibration via local clean priors is essential for bridging the representation gap caused by LDP noise. This confirms that the semantic distortion is too severe to be rectified by a frozen Head alone.

The full PISA framework, which further integrates UGR, achieves the highest performance across all settings. The contribution of UGR is most pronounced under strict privacy constraints: at $\varepsilon = 10$, UGR contributes 3.4 points on average (71.3% vs. 67.9%), with the largest gain on MNLI (68.6% vs. 62.6%). This pattern is consistent with Theorem 4.2: tighter privacy budgets generate a higher proportion of severely perturbed samples with toxic gradients, making confidence-guided gradient rescaling increasingly critical. As $\varepsilon$ increases to 14, the gap between PISA and PISA$^*$ narrows (78.1% vs. 75.7%), corroborating that gradient toxicity diminishes as perturbation intensity decreases.

## 5.4. Impact of Split Point Configuration

We investigate the impact of Tail depth ($L \in [1, 4]$) on downstream performance, as shown in Table 2. Notably, the baseline Naive-RS fails to converge even at a deep configuration ($L = 4$), whereas PISA demonstrates robustness across all depths. Specifically, PISA's utility consistently improves as $L$ increases from 1 to 3, confirming that allocating more local parameters enhances domain adaptation capacity. However, performance saturates or slightly degrades at $L = 4$ due to optimization difficulties under heavy privacy noise. Consequently, we identify $L = 3$ as the optimal sweet spot, striking a favorable balance between model capacity and client-side efficiency.

## 5.5. Effect of MRP Pre-training Fairness

A potential concern is whether PISA's gains over Hosted-LDP stem from its access to public data during MRP rather

*Table 2.* Impact of Tail depth ($L$).

| Method | Depth | SST-2 | | QNLI | |
|---|---|---|---|---|---|
| | | $\varepsilon{=}10$ | $\varepsilon{=}16$ | $\varepsilon{=}10$ | $\varepsilon{=}16$ |
| Naive-RS | $L=4$ | 51.6 | 51.6 | 50.9 | 50.9 |
| PISA | $L=1$ | 80.7 | 86.9 | 61.1 | 74.5 |
| | $L=2$ | 81.0 | 87.1 | 62.4 | 75.2 |
| | $L=3$ | **81.2** | **88.1** | **64.3** | 74.9 |
| | $L=4$ | 80.2 | 87.5 | 63.1 | **75.5** |

than its architectural design. To verify this, we equip Hosted-LDP with the same MRP pre-training on Wikipedia, denoted as Hosted-LDP+MRP. As shown in Table 3, while MRP independently improves Hosted-LDP by 16.0 points at $\varepsilon = 12$, PISA still outperforms Hosted-LDP+MRP by 9.1 points, demonstrating that the performance gains stem from the complete dual-constraint architecture rather than public data access alone. Critically, neither Hosted-LDP nor Hosted-LDP+MRP satisfies the model IP constraint, as both expose fine-tuned parameters to the server.

*Table 3.* Comparison with MRP-augmented baseline.

| Method | $\varepsilon = 12$ | $\varepsilon = 14$ |
|---|---|---|
| Hosted-LDP | 59.4 | 86.4 |
| Hosted-LDP+MRP | 75.4 | 85.2 |
| **PISA (Ours)** | **84.5** | **87.1** |

## 5.6. Effect of Public Dataset Choice

Since MRP relies on a public auxiliary dataset $\mathcal{D}_{pub}$ for offline robustness preparation, we investigate the sensitivity of PISA to the choice of public data by evaluating two datasets of contrasting domain relevance: the Wikipedia corpus and OpenWebText (Gokaslan & Cohen, 2019), under $\varepsilon = 12$ on bert-base-uncased.

*Table 4.* Effect of public dataset choice on PISA performance.

| Public Dataset | SST-2 | QNLI | MNLI |
|---|---|---|---|
| Wikipedia Corpus | 84.5 | 68.2 | 69.8 |
| OpenWebText Corpus | 82.5 | 69.4 | 71.2 |

As shown in Table 4, performance varies by at most 2.0 points across all tasks and datasets, confirming that PISA is robust to public dataset choice. This robustness stems from MRP's core objective: rather than learning task-specific semantics, $\mathcal{L}_{\text{Denoise}}$ optimizes the geometric relationship between perturbed and clean representations, acquiring a general manifold alignment capability that is not tied to any particular domain. Consequently, any sufficiently diverse

natural language corpus provides adequate coverage for effective robustness preparation.

### 5.7. Empirical Convergence Analysis of UGR

To empirically validate Theorem 4.2, we compare the training dynamics of PISA and PISA* (w/o UGR) on SST-2 under $\varepsilon = 12$. As shown in Figure 3, UGR consistently achieves faster early convergence (left) and suppresses gradient norm variance throughout training (right), confirming that confidence-guided rescaling effectively filters toxic gradients from severely perturbed samples.

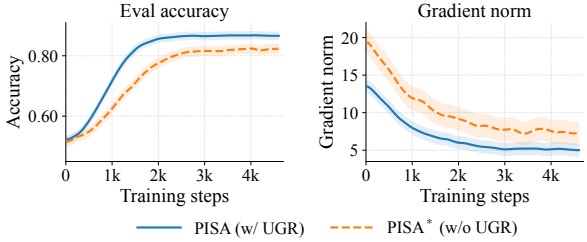

*Figure 3.* Convergence analysis of UGR.

### 5.8. Empirical Evaluation on Model Stealing

While existing SL attacks exploit backward gradients (Pasquini et al., 2021; Zhang et al., 2025), PISA's strict gradient blockage renders them structurally infeasible. However, the server receives LDP-perturbed inputs which, as shown in (Yue et al., 2021), retain sufficient utility for downstream modeling. Consequently, to evaluate IP resilience, we design a tailored extraction attack where the adversary attempts to reconstruct the proprietary model solely using the uploaded perturbed data.

**Adaptive Attack Design.** We assume an honest-but-curious server accumulates LDP-perturbed inputs $\mathcal{D}_{adv} = \{\tilde{x}_i\}_{i=1}^N$. Unlike standard attacks targeting sub-modules, the adversary leverages abundant training data to fine-tune a shadow model $\mathcal{M}_{shadow}$ replicating the complete task functionality. The primary challenge lies in the absence of ground truth labels $y_i$ for the perturbed inputs. To overcome this, we formulate two adaptive labeling strategies: 1) Attack-Rand assigns random labels to measure baseline unsupervised leakage; 2) Attack-LLM generates pseudo-labels via LLM-generator (e.g. GPT-4) on perturbed text, supervising $\mathcal{M}_{shadow}$ to approximate the data holder's decision boundary. The specific prompts used for labeling the datasets are detailed in Appendix G.

**Evaluation Protocol.** We quantify the severity of IP leakage using the Stolen Model Accuracy (SMA), defined as the classification performance of the adversary's shadow model on the test set. A high SMA comparable to the client's model indicates a breach of Model IP, while a low SMA implies

*Table 5.* Comparison of Stolen Model Accuracy (SMA).

| Method | SST-2 | | QNLI | |
|---|---|---|---|---|
| | $\varepsilon = 10$ | $\varepsilon = 14$ | $\varepsilon = 10$ | $\varepsilon = 14$ |
| **PISA (Target)** | **81.2** | **87.1** | **64.3** | **73.2** |
| Attack-Rand | 50.3 | 52.9 | 50.3 | 49.4 |
| Attack-LLM | 59.5 | 64.8 | 49.5 | 56.6 |

effective protection. We conduct experiments on two representative benchmarks under privacy budgets $\epsilon \in \{10, 14\}$. We compare the utility of PISA against the shadow models derived from Attack-Rand and Attack-LLM.

**Results and Analysis.** Table 5 presents the comparative results. First, we observe that Attack-Rand yields performance close to random guessing (e.g., $\approx 50\%$ on SST-2), confirming that merely possessing the feature distribution without task-specific supervision is insufficient to reconstruct the proprietary logic. More importantly, even under the stronger Attack-LLM, the adversary fails to replicate the client's high utility. For instance, on QNLI with $\epsilon = 14$, while PISA maintains a high accuracy of 87.1%, the stolen model only achieves 64.8%, exhibiting a significant performance gap. This degradation stems from the fact that PISA's DSC relies on local clean priors that are physically isolated from the server. The adversary, having access only to noise-corrupted embeddings $\tilde{x}$ and lacking the specific gradient guidance from the client's label $y$, cannot rectify the semantic distortion effectively. These results empirically demonstrate that PISA breaks the trade-off between utility and security, providing a robust guarantee against IP theft.

Furthermore, we empirically validate the data privacy guarantees of PISA in Appendix H.

## 6. Conclusion

In this work, we present PISA, a split fine-tuning framework designed to simultaneously uphold strict data privacy and model IP protection. By integrating prior-guided semantic restoration mechanisms, PISA effectively resolves the "Frozen-Head, Limited-Tail" bottleneck inherent in dual-constraint collaborative architectures. Our results demonstrate that PISA delivers utility comparable to single-objective baselines, successfully bridging the gap between privacy compliance and model sovereignty without compromising performance. Future work will focus on extending PISA to generative tasks, which requires dedicated mechanism design as DSC's compensation cannot propagate to newly generated tokens during autoregressive decoding. Adapting PISA for on-device deployment in resource-constrained scenarios also remains an open direction.

## Acknowledgements

This work was supported in part by the National Natural Science Foundation of China (NSFC) under Grant No. 62372051, and in part by the Key Laboratory of Computing Power Network and Information Security, Ministry of Education, under Grant No. 2024PY010.

## Impact Statement

This paper presents work aimed at advancing the field of secure collaborative fine-tuning for large language models. We introduce the PISA framework, which integrates Manifold Rectification Pre-training (MRP) method, Dual-Stream Semantic Compensation (DSC) method, and Utility-Aware Gradient Rectification (UGR) method to simultaneously protect data privacy and model intellectual property. To the best of our knowledge, we have not identified any negative effects associated with our research that merit highlighting in this discussion.

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

# A. Privacy Analysis

In this section, we provide a rigorous analysis of the privacy guarantees inherent to the PISA framework. The privacy preservation strategy is structurally decoupled according to the two operational phases of the framework.

First, regarding the offline robustness preparation phase, the server conducts the Manifold Rectification Pre-training (MRP) exclusively utilizing a public auxiliary dataset $\mathcal{D}_{pub}$. Since this process occurs entirely within the server's domain and involves no interaction with the data holder or their private dataset $\mathcal{D}_C$, this phase is theoretically immune to privacy leakage regarding the client's sensitive information. The resultant foundation model parameters are derived solely from public knowledge, ensuring that the initialization of the Head $F_\phi$ and Tail $G_\theta$ contains no traces of private data.

Second, regarding the online collaborative fine-tuning phase, the interaction involves the transmission of data from the data holder to the server. To neutralize the risk of data reconstruction or membership inference, the data holder applies a sanitized mechanism $\mathcal{M}$ to the raw input $x$ before transmission. Specifically, we employ the SanText mechanism (Yue et al., 2021), which is mathematically proven to satisfy $\epsilon d_\chi$-privacy. Crucially, PISA is mechanism-agnostic. The framework strictly adheres to the privacy bounds of the employed noise mechanism, ensuring that the overall protection level is equivalent to that of the specific instantiation regardless of the mechanism selection. Consequently, the server's view is strictly limited to the perturbed representation $\tilde{x} = \mathcal{M}(x)$. Furthermore, the reverse split architecture combined with the gradient blockage protocol ensures that the server receives no backward gradient information that could carry derivatives of the private labels $y$. Therefore, the privacy guarantee of the entire PISA framework in the collaborative phase reduces to the privacy guarantee of the input sanitization mechanism $\mathcal{M}$.

We formally prove that the sanitization mechanism utilized in PISA satisfies the definition of $\epsilon d_\chi$-privacy. The mechanism $\mathcal{M}$ operates by sampling a perturbed token $y$ from the vocabulary $\mathcal{V}$ with a probability proportional to the exponential of the negative Euclidean distance between the input embedding and the candidate embedding. Specifically, for an input token $x$, the probability of outputting $y$ is given by $Pr[\mathcal{M}(x) = y] = \frac{1}{C_x} \exp\left(-\frac{\epsilon}{2} d(x, y)\right)$, where $d(x, y)$ represents the Euclidean distance between the embeddings of $x$ and $y$, and $C_x = \sum_{v \in \mathcal{V}} \exp\left(-\frac{\epsilon}{2} d(x, v)\right)$ is the normalization constant.

**Theorem A.1.** *The PISA framework satisfies $\epsilon d_\chi$-privacy.*

*Proof.* Consider an arbitrary pair of input tokens $x, x' \in \mathcal{X}$ and a potential output token $y \in \mathcal{Y}$. We examine the log-likelihood ratio of the mechanism outputting $y$ given inputs $x$ and $x'$:

$$\frac{Pr[\mathcal{M}(x) = y]}{Pr[\mathcal{M}(x') = y]} = \frac{\frac{1}{C_x} \exp\left(-\frac{\epsilon}{2} d(x, y)\right)}{\frac{1}{C_{x'}} \exp\left(-\frac{\epsilon}{2} d(x', y)\right)} = \frac{C_{x'}}{C_x} \exp\left(\frac{\epsilon}{2}(d(x', y) - d(x, y))\right) \quad (12)$$

Invoking the triangle inequality of the metric $d$, we have $d(x', y) \leq d(x', x) + d(x, y)$, which implies $d(x', y) - d(x, y) \leq d(x, x')$. Substituting this inequality into the exponential term yields:

$$\frac{Pr[\mathcal{M}(x) = y]}{Pr[\mathcal{M}(x') = y]} \leq \frac{C_{x'}}{C_x} \exp\left(\frac{\epsilon}{2} d(x, x')\right) \quad (13)$$

We next bound the ratio of the normalization constants $\frac{C_{x'}}{C_x}$. By definition, $C_{x'} = \sum_{v \in \mathcal{V}} \exp\left(-\frac{\epsilon}{2} d(x', v)\right)$. Applying the triangle inequality $d(x', v) \geq d(x, v) - d(x, x')$, we derive:

$$C_{x'} \leq \sum_{v \in \mathcal{V}} \exp\left(-\frac{\epsilon}{2}(d(x, v) - d(x, x'))\right) = \exp\left(\frac{\epsilon}{2} d(x, x')\right) \sum_{v \in \mathcal{V}} \exp\left(-\frac{\epsilon}{2} d(x, v)\right) \quad (14)$$

Observing that the summation term is exactly $C_x$, we obtain the bound $\frac{C_{x'}}{C_x} \leq \exp\left(\frac{\epsilon}{2} d(x, x')\right)$. Combining this result with the previous inequality provides the final bound:

$$\frac{Pr[\mathcal{M}(x) = y]}{Pr[\mathcal{M}(x') = y]} \leq \exp\left(\frac{\epsilon}{2} d(x, x')\right) \cdot \exp\left(\frac{\epsilon}{2} d(x, x')\right) = e^{\epsilon \cdot d(x, x')} \quad (15)$$

This derivation confirms that the token-level sanitization satisfies $\epsilon d_\chi$-privacy. For a sequence of tokens $D = \{x_1, \ldots, x_L\}$, the mechanism applies independent sanitization to each token. According to the sequential composition property of differential privacy, the privacy guarantee for the entire sequence extends naturally. Specifically, for two sequences $D$ and $D'$ differing at each position, the joint probability ratio is bounded by the product of individual bounds, yielding strict adherence to the $d_\chi$-privacy definition over the sequence domain. $\qquad \square$

## B. Theoretical Complexity Analysis

In this section, we provide a formal complexity analysis of PISA compared to Standard Split Learning (SL) and Local Full Fine-tuning (Local FT). We prioritize the analysis of client-side computational overhead, as the edge device's processing capability typically constitutes the primary bottleneck in collaborative adaptation, whereas the server is assumed to possess abundant computational resources. We define the complexity in terms of Computation ($\mathcal{T}$), Communication ($\mathcal{C}$), and Memory ($\mathcal{M}$) per training step.

**Notations.** Let $B$, $S$, and $D$ denote the batch size, sequence length, and hidden dimension, respectively. Let $L$ be the total number of transformer layers in the foundation model, partitioned into $L_{head}$ server-side layers and $L_{tail}$ client-side layers, such that $L = L_{head} + L_{tail}$. We denote $\mathcal{F}$ as the floating-point operations (FLOPs) required for the forward pass of a single layer. Following standard estimation, the backward pass cost is approximated as $\mathcal{B} \approx 2\mathcal{F}$.

### B.1. Computational Complexity

We first examine the computational burden placed on the data holder. In the standard Local Full Fine-tuning paradigm, the client is required to execute both forward and backward propagation for the entire model, resulting in a total computational cost of $\mathcal{T}_{Local} \approx L(\mathcal{F} + \mathcal{B}) \approx 3L \cdot \mathcal{F}$. In contrast, the PISA framework structurally offloads the computation-intensive Head to the server, restricting the client's burden solely to the lightweight Tail sub-model. Consequently, the local computational complexity is reduced to $\mathcal{T}_{PISA} \approx L_{tail}(\mathcal{F} + \mathcal{B}) \approx 3L_{tail} \cdot \mathcal{F}$. Given that $L_{tail} \ll L$ in our asymmetric design, this optimization provides a significant reduction in FLOPs by a factor of $L/L_{tail}$, effectively enabling efficient adaptation on resource-constrained devices.

### B.2. Communication Complexity

Regarding communication overhead, Standard Split Learning necessitates a bidirectional exchange of intermediate activations and gradient signals at the cut layer, yielding a complexity of $\mathcal{C}_{SL} = 2 \cdot B \cdot S \cdot D$ per step. Conversely, PISA enforces a strict gradient blockage protocol which eliminates the backward transmission of gradients. The client only receives the perturbed features during the forward phase, thereby halving the bandwidth consumption to $\mathcal{C}_{PISA} = 1 \cdot B \cdot S \cdot D$. This unidirectional flow significantly mitigates synchronization latency compared to conventional collaborative paradigms.

### B.3. Memory Complexity

Finally, we analyze the memory footprint, which is dominated by model parameters $\Theta$, optimizer states $\Omega$, and activation maps $\mathcal{A}$. While local fine-tuning requires storing these components for the full foundation model, PISA strictly decouples the local memory requirement from the global model scale. The client maintains states exclusively for the Tail module, resulting in a memory complexity of $\mathcal{M}_{PISA} \approx \Theta_{tail} + \Omega_{tail} + \mathcal{A}_{tail}$. Since the parameter count of the Tail is constant and independent of the server-side Head, PISA effectively overcomes the memory wall, allowing consumer-grade hardware to participate in the fine-tuning of large-scale models regardless of the foundation model size.

## C. Proof of Theorem 4.2

In this section, we provide the detailed proof for the convergence analysis of the Utility-Aware Gradient Rectification (UGR) mechanism. Unlike simplified analyses that assume independence between quality scores and gradient noise, we present a rigorous derivation under a relaxed assumption of bounded correlation.

### C.1. Setup and Assumptions

Let $f(\theta) := \mathbb{E}[\mathcal{L}_{task}(\theta)]$ denote the expected loss function. The update rule at step $t$ is $\theta_{t+1} = \theta_t - \eta\tilde{g}_t$, where $\tilde{g}_t = \alpha_t g(\theta_t)$ is the rectified stochastic gradient.

We rely on the following assumptions, consistent with Assumption 4.1 in the main text:

1. L-Smoothness: The objective function $f$ is $L$-smooth, satisfying:

$$f(\theta_{t+1}) \leq f(\theta_t) + \langle \nabla f(\theta_t), \theta_{t+1} - \theta_t \rangle + \frac{L}{2}\|\theta_{t+1} - \theta_t\|^2 \tag{16}$$

2. Unbiasedness & Bounded Correlation: The stochastic gradient is unbiased, i.e., $\mathbb{E}[g(\theta_t)|\theta_t] = \nabla f(\theta_t)$. However, we acknowledge that the quality score $\alpha_t$ and the stochastic gradient $g(\theta_t)$ originate from the same perturbed input, introducing a statistical dependence. We assume this correlation is bounded. Specifically, let the expectation of the rectified gradient be:

$$\mathbb{E}_t[\alpha_t g(\theta_t)] = \mathbb{E}_t[\alpha_t]\nabla f(\theta_t) + \xi_t \tag{17}$$

where $\xi_t$ represents the correlation bias vector. We assume there exists a constant $\delta \geq 0$ such that $\|\xi_t\| \leq \delta$. Note that standard independence assumptions correspond to the case where $\delta = 0$.

For the second moment, we adopt the standard decomposition:

$$\mathbb{E}_t[\alpha_t^2\|g(\theta_t)\|^2] \leq \mathbb{E}_t[\alpha_t^2](\|\nabla f(\theta_t)\|^2 + \sigma_{noise}^2) \tag{18}$$

3. Bounded Variance: The variance of the stochastic gradient is bounded by the privacy noise level $\sigma_{noise}^2$.

## C.2. Proof Derivation

Substituting the UGR update rule $\theta_{t+1} - \theta_t = -\eta\alpha_t g(\theta_t)$ into the smoothness inequality:

$$f(\theta_{t+1}) \leq f(\theta_t) - \eta\langle\nabla f(\theta_t), \alpha_t g(\theta_t)\rangle + \frac{L\eta^2}{2}\alpha_t^2\|g(\theta_t)\|^2 \tag{19}$$

Taking the conditional expectation $\mathbb{E}_t = \mathbb{E}[\cdot|\theta_t]$:

1. Analysis of the Linear Term (with Correlation Bias): Substituting the bounded correlation assumption:

$$\begin{aligned}-\eta\langle\nabla f(\theta_t), \mathbb{E}_t[\alpha_t g(\theta_t)]\rangle &= -\eta\langle\nabla f(\theta_t), \mathbb{E}_t[\alpha_t]\nabla f(\theta_t) + \xi_t\rangle \\ &= -\eta\mathbb{E}_t[\alpha_t]\|\nabla f(\theta_t)\|^2 - \eta\langle\nabla f(\theta_t), \xi_t\rangle\end{aligned} \tag{20}$$

To bound the bias term $-\eta\langle\nabla f(\theta_t), \xi_t\rangle$, we employ Young's Inequality with a scaling factor $\mathbb{E}_t[\alpha_t] > 0$:

$$-\eta\langle\nabla f(\theta_t), \xi_t\rangle \leq \eta\|\nabla f(\theta_t)\|\|\xi_t\| \leq \frac{\eta\mathbb{E}_t[\alpha_t]}{2}\|\nabla f(\theta_t)\|^2 + \frac{\eta}{2\mathbb{E}_t[\alpha_t]}\|\xi_t\|^2 \tag{21}$$

Substituting $\|\xi_t\| \leq \delta$, the linear term is bounded by:

$$-\eta\langle\nabla f(\theta_t), \mathbb{E}_t[\alpha_t g(\theta_t)]\rangle \leq -\frac{\eta\mathbb{E}_t[\alpha_t]}{2}\|\nabla f(\theta_t)\|^2 + \frac{\eta\delta^2}{2\mathbb{E}_t[\alpha_t]} \tag{22}$$

This step is crucial as it demonstrates that despite the bias, the gradient descent direction is preserved (halving the effective descent coefficient) at the cost of an additive error term.

2. Analysis of the Quadratic Term: Using the bounded variance assumption:

$$\frac{L\eta^2}{2}\mathbb{E}_t[\alpha_t^2\|g(\theta_t)\|^2] \leq \frac{L\eta^2}{2}\mathbb{E}_t[\alpha_t^2](\|\nabla f(\theta_t)\|^2 + \sigma_{noise}^2) \tag{23}$$

3. Combined Inequality: Combining the linear and quadratic bounds:

$$\begin{aligned}\mathbb{E}_t[f(\theta_{t+1})] \leq f(\theta_t) &- \frac{\eta\mathbb{E}_t[\alpha_t]}{2}\|\nabla f(\theta_t)\|^2 + \frac{\eta\delta^2}{2\mathbb{E}_t[\alpha_t]} \\ &+ \frac{L\eta^2}{2}\mathbb{E}_t[\alpha_t^2]\|\nabla f(\theta_t)\|^2 + \frac{L\eta^2}{2}\mathbb{E}_t[\alpha_t^2]\sigma_{noise}^2\end{aligned} \tag{24}$$

Rearranging to isolate the gradient norm $\|\nabla f(\theta_t)\|^2$:

$$\mathbb{E}_t[f(\theta_{t+1})] \leq f(\theta_t) - \frac{\eta}{2}\left(\mathbb{E}_t[\alpha_t] - L\eta\mathbb{E}_t[\alpha_t^2]\right)\|\nabla f(\theta_t)\|^2 + \frac{L\eta^2}{2}\mathbb{E}_t[\alpha_t^2]\sigma_{noise}^2 + \frac{\eta\delta^2}{2\mathbb{E}_t[\alpha_t]} \tag{25}$$

Since $\alpha_t \in [0, 1]$, we have $\alpha_t^2 \le \alpha_t$. Thus, $\mathbb{E}_t[\alpha_t] - L\eta\mathbb{E}_t[\alpha_t^2] \ge \mathbb{E}_t[\alpha_t](1 - L\eta)$. Choosing a step size $\eta \le \frac{1}{2L}$ ensures $(1 - L\eta) \ge \frac{1}{2}$. The inequality simplifies to:

$$\mathbb{E}_t[f(\theta_{t+1})] \le f(\theta_t) - \frac{\eta}{4}\mathbb{E}_t[\alpha_t]\|\nabla f(\theta_t)\|^2 + \frac{L\eta^2}{2}\mathbb{E}_t[\alpha_t^2]\sigma_{noise}^2 + \frac{\eta\delta^2}{2\mathbb{E}_t[\alpha_t]} \tag{26}$$

Rearranging to bound the weighted gradient norm:

$$\frac{\eta}{4}\mathbb{E}_t[\alpha_t]\|\nabla f(\theta_t)\|^2 \le f(\theta_t) - \mathbb{E}_t[f(\theta_{t+1})] + \frac{L\eta^2}{2}\mathbb{E}_t[\alpha_t^2]\sigma_{noise}^2 + \frac{\eta\delta^2}{2\mathbb{E}_t[\alpha_t]} \tag{27}$$

Taking the total expectation and summing over $t = 1$ to $T$:

$$\frac{\eta}{4}\sum_{t=1}^{T}\mathbb{E}[\alpha_t\|\nabla f(\theta_t)\|^2] \le f(\theta_1) - f^* + \frac{L\eta^2}{2}\sum_{t=1}^{T}\mathbb{E}[\alpha_t^2]\sigma_{noise}^2 + \sum_{t=1}^{T}\frac{\eta\delta^2}{2\mathbb{E}[\alpha_t]} \tag{28}$$

Dividing by $\frac{\eta}{4}A_T$ where $A_T = \sum_{t=1}^{T}\mathbb{E}[\alpha_t]$, and defining $\mathcal{G}_T^2$ as the weighted average gradient norm:

$$\mathcal{G}_T^2 \le \frac{4(f(\theta_1) - f^*)}{\eta A_T} + \frac{2L\eta}{A_T}\sum_{t=1}^{T}\mathbb{E}[\alpha_t^2]\sigma_{noise}^2 + \frac{2\delta^2}{A_T}\sum_{t=1}^{T}\frac{1}{\mathbb{E}[\alpha_t]} \tag{29}$$

Finally, substituting $\eta = \frac{1}{\sqrt{T}}$ and defining the average quality $\bar{\alpha} = A_T/T$:

$$\mathcal{G}_T^2 \le \frac{4(f(\theta_1) - f^*)}{\bar{\alpha}\sqrt{T}} + \frac{2L}{\sqrt{T}}\left(\frac{1}{T\bar{\alpha}}\sum_{t=1}^{T}\mathbb{E}[\alpha_t^2]\sigma_{noise}^2\right) + \mathcal{O}(\delta^2) \tag{30}$$

This confirms that UGR converges with a rate of $\mathcal{O}(1/\sqrt{T})$ to a neighborhood determined by the correlation bias $\delta^2$. As the privacy budget $\epsilon$ increases (reducing perturbation), the correlation $\delta$ naturally diminishes, recovering the standard convergence properties.

## D. Datasets

The statistical characteristics of the datasets are presented in Table 6.

*Table 6.* Statistics of the datasets.

| Dataset | SST-2 | QNLI | MNLI |
|---|---|---|---|
| Train Samples | 67,350 | 104,744 | 392,702 |
| Test Samples | 873 | 5,462 | 19,647 |

## E. Detailed Implementation Settings

In this section, we provide the comprehensive experimental configurations to ensure reproducibility. Our framework is implemented using PyTorch and the HuggingFace Transformers library.

### E.1. Model Architecture and Splitting

We utilize BERT-base-uncased as the backbone models. For the reverse split architecture, we partition the model at the 3 layer from the top. Consequently, the server-side Head consists of the bottom 9 layers, while the client-side Tail comprises the remaining upper layers and the classification head. The maximum sequence length is set to 128 for all tasks.

### E.2. Phase I: Manifold Rectification Pre-training

The offline robustness preparation is conducted on the server using the English Wikipedia dump. We preprocess the data to exclude headers and verify that it contains no overlap with the downstream GLUE benchmarks. During this phase, the model is trained for 5 epochs with a batch size of 32 to align the feature manifold.

### E.3. Phase II: Collaborative Fine-tuning

For the online adaptation phase, we employ the AdamW optimizer with a linear learning rate decay scheduler. We perform a grid search for the optimal learning rate within the range of 1e-5 and set the batch size to 16. The privacy budget $\epsilon$ for the LDP mechanism varies from 6 to 18 to evaluate performance under different protection levels. All experiments are repeated 5 times with different random seeds, and we report the average performance.

## F. Efficiency Comparison Results

To demonstrate the deployment feasibility of PISA on resource-constrained edge devices, we conducted a comprehensive efficiency evaluation using two representative backbones: BERT-base and Llama2-7B. We benchmark PISA against two baselines: Local Full Fine-tuning (Local FT), representing the resource-intensive standard adaptation, and Standard Split Learning (Standard SL), representing the conventional collaborative adaptation with bidirectional gradient transmission.

We measured four critical metrics: peak client-side GPU memory usage (VRAM), total communication traffic, training throughput, and total convergence time. For the experimental setup, we configured the split point to offload the majority of the computational burden to the server. Specifically, for the 70-layer Llama2-7B, the bottom 24 layers are frozen on the server, leaving only 8 layers on the client; for the 12-layer BERT-base, the bottom 9 layers are hosted on the server, leaving the top 3 layers on the client.

Table 7 presents the comparative results. First, regarding memory efficiency, Local FT imposes prohibitive hardware demands for the large-scale Llama2-7B model, resulting in Out-Of-Memory (OOM) errors on consumer-grade GPUs such as the RTX 3090. In contrast, PISA effectively breaks this memory wall by reducing the footprint by approximately 72.4%, making LLM adaptation feasible on a single consumer-grade card.

Moreover, PISA demonstrates significant advantages in communication efficiency. Unlike Standard SL which suffers from high latency due to the bidirectional transmission of activations and gradients, the unidirectional forward-only protocol of PISA reduces communication overhead by exactly 50%, thereby alleviating the bandwidth bottleneck inherent in collaborative learning.

Notably, despite the introduction of network latency, PISA achieves higher training throughput than Local FT, such as a 1.5x speedup for Llama2-7B. This counter-intuitive acceleration is attributed to the frozen head design, where eliminating the computationally expensive backward propagation for the massive server-side parameters generates computational savings that far outweigh the additional communication latency. This confirms that PISA not only protects IP but also serves as an efficient acceleration framework.

*Table 7.* Computational efficiency comparison on QNLI. Standard SL values are estimated based on the bidirectional gradient transmission protocol. OOM indicates Out-Of-Memory on a 24GB NVIDIA RTX 3090.

| Model | Method | Client VRAM (MB) ↓ | Total Comm. (MB) ↓ | Throughput (samples/s) ↑ | Total Time (min) ↓ |
|---|---|---|---|---|---|
| **BERT-base** | Local FT | 3,776 | **0** | 136.0 | 6 |
| | Standard SL[†] | 1,830 | 52,974 | 85.5 | 10 |
| | **PISA (Ours)** | **1,828** | **26,487** | **219.4** | **3** |
| **Llama2-7B** | Local FT | 54,833 (OOM) | **0** | 8.9 | 77 |
| | Standard SL[†] | 15,200 | 69,222 | 4.2 | 165 |
| | **PISA (Ours)** | **15,151** | **34,611** | **13.6** | **19** |

[†] Standard SL results are projected based on PISA architecture with gradient sharing enabled.

# G. Prompts for Adaptive Attack

In the Attack-LLM strategy described in Section 5.8, we utilize an LLM (e.g., GPT-4) to generate pseudo-labels for the perturbed text. The specific system prompts designed for the SST-2 and QNLI datasets are presented in Table 8. These prompts guide the LLM to act as a domain-specific classifier, adhering to strict output constraints to ensure compatibility with the shadow model training.

*Table 8.* System prompts used for LLM-based pseudo-labeling (Attack-LLM) on SST-2 and QNLI datasets.

| Dataset | System Prompt Content |
|---|---|
| **SST-2** | Role: SST-2 sentiment labeler.
Decide the overall sentiment of the sentence.

Decision guide:
1) If the sentence expresses praise, enjoyment, recommendation, or strong approval → `positive`.
2) If it expresses criticism, disappointment, boredom, dislike, or strong disapproval → `negative`.
3) If mixed, choose the dominant sentiment.

Output constraints:
- First line must be exactly: `positive` OR `negative` (lowercase).
- Second line is optional: $\leq 10$ words reason. If unsure, omit the second line. |
| **QNLI** | Role: QNLI classifier.
Given a Question and a Sentence, decide if the Sentence answers the Question.

Decision guide:
1) `entailment`: the Sentence contains the answer or clearly implies it.
2) `not_entailment`: missing the answer, irrelevant, too vague, or only loosely related.
3) If uncertain, choose `not_entailment`.

Output constraints:
- First line must be exactly: `entailment` OR `not_entailment` (lowercase).
- No other labels (no neutral / contradiction). |

# H. Data Privacy Evaluation against Inference Attacks

While the primary focus of PISA is to introduce Model IP protection into collaborative adaptation, it is imperative to verify that our split architecture does not compromise the fundamental data privacy guarantees provided by the underlying Local Differential Privacy (LDP) mechanism. To this end, we conduct a Mask Token Inference Attack to evaluate the resistance of PISA against textual reconstruction, comparing it directly with the standalone baseline LDP mechanism. This comparative analysis aims to demonstrate that incorporating the proposed split fine-tuning framework does not introduce additional vulnerabilities or degrade the intrinsic defense capabilities of the privacy module.

## H.1. Experimental Setup

We adhere to the standard threat model where an honest-but-curious server attempts to reconstruct the original sensitive tokens from the sanitized inputs. We employ a pre-trained BERT model as the adversary, which utilizes the bidirectional context of the sanitized text to predict the original tokens masked sequentially. We measure the Defense Rate, defined as the percentage of tokens that the adversary fails to recover correctly. We compare the defense performance of our PISA framework against the Base LDP mechanism (operating in a standard non-split setting) on the SST-2 and QNLI datasets under varying privacy budgets $\varepsilon$.

## H.2. Comparative Results

The comparative results are presented in Table 9. We observe that the defense rates of PISA are statistically consistent with those of the Base LDP mechanism across all privacy budgets. For instance, at $\varepsilon = 10$ on the SST-2 dataset, PISA achieves a defense rate of 94.2%, which is comparable to the 94.0% exhibited by the Base LDP. This alignment confirms that the privacy protection in PISA is strictly governed by the configured $\varepsilon$ of the LDP module and is not negatively impacted by the subsequent split adaptation process. Furthermore, as $\varepsilon$ increases (implying reduced noise), the defense rates for both methods decrease in a synchronized manner, adhering to the theoretical privacy-utility trade-off. These findings empirically validate that PISA successfully preserves the rigorous data privacy standards of the underlying LDP mechanism while enabling secure collaborative fine-tuning.

*Table 9.* Comparative Defense Rate (%) against Mask Token Inference Attack on SST-2 and QNLI datasets. "Base LDP" represents the standalone local differential privacy mechanism, and "PISA" represents our proposed framework. Higher Defense Rate indicates stronger protection against reconstruction.

| Dataset | $\varepsilon = 10$ | | $\varepsilon = 12$ | | $\varepsilon = 14$ | | $\varepsilon = 16$ | |
|---|---|---|---|---|---|---|---|---|
| | Base LDP | **PISA** | Base LDP | **PISA** | Base LDP | **PISA** | Base LDP | **PISA** |
| SST-2 | 94.0 | 94.2 | 89.3 | 89.5 | 81.5 | 81.3 | 72.8 | 72.6 |
| QNLI | 95.0 | 95.1 | 91.9 | 91.8 | 84.6 | 84.7 | 75.5 | 75.4 |

# I. Extension to Decoder-Only Architectures

To validate the generalizability of PISA beyond encoder-only architectures, we evaluate the framework on Llama2-7B under the same dual-constraint setting. as described at the end of Section 4.4, $\mathcal{L}_{\text{MLM}}$ is replaced by $\mathcal{L}_{\text{CLM}}$, and the confidence score $\alpha$ is computed as the sequence-level average log-likelihood. The model is split at layer 24, designating the bottom 24 layers as the server Head and the upper 8 layers plus the classification head as the client Tail. Results are averaged over 3 runs.

*Table 10.* Performance of PISA on Llama2-7B (split = 24 Head / 8 Tail layers).

| Method | SST-2 | | QNLI | |
|---|---|---|---|---|
| | $\varepsilon = 12$ | $\varepsilon = 16$ | $\varepsilon = 12$ | $\varepsilon = 16$ |
| Vanilla FT (Non-DP) | 93.5 | 93.5 | 92.5 | 92.5 |
| Naive-RS | 50.4 | 50.4 | 51.1 | 51.1 |
| Hosted-LDP | 68.7 | 90.6 | 62.2 | 88.7 |
| **PISA (Ours)** | **75.6** | **86.2** | **65.0** | **75.2** |

Under strict privacy ($\varepsilon = 12$), PISA consistently outperforms Hosted-LDP on both tasks, while Naive-RS fails to converge. The crossover at $\varepsilon = 16$ mirrors the pattern in Figure 2 and is expected: relaxed noise allows Hosted-LDP's full-parameter advantage to emerge, at the cost of model IP exposure. These results confirm that PISA's three components transfer effectively to decoder-only architectures without architectural modification.

