# OpenReview forum: "PISA: Privacy-Preserving Split Adaptation with Model IP Protection"
_ICML.cc/2026/Conference — ICML 2026 regular_

### Official Review · Reviewer_rQ4D · 2026-02-21

**Soundness:** 3
**Presentation:** 4
**Significance:** 3
**Originality:** 2
**Overall Recommendation:** 5
**Confidence:** 4

**Summary:**

This paper introduces **PISA**, designed for collaborative LLM fine-tuning between resource-rich server and private-data holder. The authors identify fundamental conflicts, data privacy and model Intellectual Property (IP) during this process.  To address these limitations, PISA employs a dual-stream fine-tuning strategy that splits the foundation model into **Head** and **Tail** components, supported by three methods: (1) **Manifold Rectification Pre-training (MRP)**, which equips the server-side backbone with noise-robustness; (2) **Dual-Stream Semantic Compensation (DSC)**, which integrates the server’s comprehensive but noisy features with the data holder’s clean but limited features; and (3) **Utility-Aware Gradient Rectification (UGR)**, which dynamically calibrates gradient updates based on confidence scores. Finally, the authors provide theoretical convergence bounds and extensive empirical evidences to demonstrate PISA’s effectiveness in terms of performance, privacy, and IP protection.

**Compliance With Llm Reviewing Policy:**

Affirmed.

**Final Justification:**

After carefully reading the authors' rebuttal and the other reviews, I maintain my score to Accept.

**Strengths:**
This paper directly targets FL-LLM environment and identifies existing limitations in simultaneously addressing data privacy and model intellectual property (IP) protection. The motivation behind PISA is compelling, and the proposed Head/Tail split strategy along with MRP, DSC, UGR are well-justified. In addition, authors provide extensive experiments covering performance evaluation, adversarial attack scenarios, and ablation studies.

**Weaknesses:**
My major concern was regarding the justification for leveraging publicly available datasets and fairness of experiments.

**Post-rebuttal:**
The authors' rebuttal adequately addressed my main concerns (Weakness 1 & Q1).

**Key Questions For Authors:**

1. On Table.1, could the authors clarify if the baselines also leveraged public dataset as PISA does? Although baselines suffer from information bottlenecks, this is essential to ensure a fair comparison, especially in high-privacy budget scenario (e.g., \epsilon=14), where Hosted-LDP demonstrates comparable or exceeding performance, that of PISA.

2. Does MLM stands for Masked Large Language model? This term first appears in Section 4.2 and Figure.1, yet its full definition is not provided. To assist readers who may be less familiar with this field, it would be helpful to explicitly define the acronym somewhere.

3. While UGR is introduced to mitigate training instability caused by perturbed inputs, its core mechanism appears to function similarly to conventional learning rate schedulers of adaptive optimizers. To better articulate the authors' contribution, it would be valuable to provide a clearer distinction between UGR and conventional optimization methods.

**Limitations:**

Yes.

**Strengths And Weaknesses:**

Strengths : This paper is well-written and logically organized, ensuring easy to understand a flow of ideas. Also, theoretical claims and empirical results are consistently aligned, supporting author’s proposed method.

Weaknesses : There is a concern regarding the reliance on strong priors derived from public dataset to obtain a noise-robust backbone.  It remains unclear whether such approach ensures a fair comparison against baselines that do not utilize such priors. Furthermore, the extent to which these public datasets can generalize to diverse, real-world private data remains questionable.

---

> ### Author Rebuttal · Authors · 2026-03-31
>
> Thank you for your feedback and valuable questions.
>
> **Weakness 1 & Q1 (Fairness of Baseline Comparison under Public Data Access).**
>
> Among our baselines, Vanilla FT does not apply LDP and has no distribution shift to correct, making MRP inapplicable. Naive-RS applies LDP but fails to converge under gradient blockage regardless of initialization, making MRP pre-training uninformative in that setting. Hosted-LDP applies LDP with full-parameter server-side fine-tuning, making it the only meaningful baseline for this comparison. To directly verify fairness, we equip Hosted-LDP with the same MRP pre-training on Wikipedia (BERT-base, SST-2):
>
> | Method         | Public Pre-training | ε=12 | ε=14 |
> | -------------- | ------------------- | ---- | ---- |
> | Hosted-LDP     | No                  | 59.4 | 86.4 |
> | Hosted-LDP+MRP | Yes                 | 75.4 | 85.2 |
> | PISA           | Yes                 | 84.5 | 87.1 |
>
> MRP alone brings Hosted-LDP from 59.4 to 75.4 (+16.0) at ε=12, confirming that manifold alignment is independently effective. PISA consistently outperforms both, and neither Hosted-LDP variant satisfies our model IP constraint. Regarding ε=14 where Hosted-LDP approaches PISA, this is consistent with Section 5.2: as noise decreases, the full-parameter fine-tuning advantage of Hosted-LDP emerges at the cost of model IP exposure. We will include Hosted-LDP+MRP as an additional baseline in the camera-ready version.
>
> Regarding generalization to diverse private data: by design, MRP minimizes $\mathcal{L}\_{\text{Denoise}}(h\_{\text{pred}},
> h\_{\text{clean}})$, which optimizes the geometric relationship between  perturbed and clean representations rather than task-specific semantics. MRP therefore learns a general manifold alignment capability independent of the public dataset's domain. We verify this empirically with two datasets of contrasting relevance (ε=12):
>
> | Public Data        | SST-2 | QNLI | MNLI |
> | ------------------ | ----- | ---- | ---- |
> | Wikipedia Corpus   | 84.5  | 68.2 | 69.8 |
> | OpenWebText Corpus | 82.5  | 69.4 | 71.2 |
>
> Performance varies by at most 2.0 points, confirming that PISA is robust to public dataset choice and generalizes to diverse private data domains.
>
> **Q2 (MLM Definition).**
>
> We will explicitly define "Masked Language Modeling (MLM)" at its first occurrence in Section 4.2. We thank the reviewer for catching this.
>
> **Q3 (UGR vs. Adaptive Optimizers).**
>
> UGR and adaptive optimizers (e.g., Adam) are fundamentally distinct and complementary. UGR operates at the sample level before gradient aggregation, acting as a quality filter that down-weights gradients from semantically corrupted inputs based on the server's linguistic confidence. Adam operates after gradient aggregation at the parameter level, adapting step sizes based on historical gradient momentum. The two mechanisms address different sources of optimization instability and are already used jointly in our experiments, which employ AdamW alongside UGR.
>
> At last, we sincerely appreciate your valuable feedback, and we will carefully consider all your suggestions to further improve our paper. Thank you very much!

---

> > ### Author Rebuttal · Reviewer_rQ4D · 2026-04-01
> >
> > My major concern was the fairness of experiments (W1, Q1). However, authors clearly verified that PISA also outperforms baselines  with same public dataset and can generalize well to diverse client dataset.
> >
> > I raised my score.

---

> > > ### Author Response · Authors · 2026-04-02
> > >
> > > Thank you for your positive feedback and for raising your score. We are glad the clarifications were helpful and will incorporate all suggested improvements in the revision.

---

### Official Review · Reviewer_2NiV · 2026-03-03

**Soundness:** 3
**Presentation:** 3
**Significance:** 4
**Originality:** 3
**Overall Recommendation:** 5
**Confidence:** 4

**Summary:**

The paper proposes a split fine-tuning framework named PISA, which aims to simultaneously preserve data and model privacy. The framework consists of three modules designed to protect privacy while minimizing loss of model utility. Experimental results show that the proposed framework yields 23% performance gain compared to existing privacy-preserving baselines.

**Compliance With Llm Reviewing Policy:**

Affirmed.

**Final Justification:**

The rebuttal has addressed my main concerns. Therefore, I changed my overall recommendation from weak accept to accept.

**Key Questions For Authors:**

1. Can the proposed framework be applied to tasks with different modalities?
2. How would the performance be affected if models with different structures are used?
3. How would the performance be affected for various public datasets (e.g., with different tasks/modalities, or different levels of relevance with the private client dataset)?

**Limitations:**

Please see the weaknesses and key questions above for potential improvements.

**Strengths And Weaknesses:**

Strengths:
- The paper is easy to follow. The motivations and the definition of the threat model are clear.
- The three modules of the framework are clearly defined and justified with both theoretical and experimental analysis.

Weaknesses:
- Analysis focuses only on linguistic tasks and is based on the assumption that the server and all clients share the same type of task.
- Experiments analyse only one model.
- Various public server datasets might have different amounts of information shared with the clients’ private data. However, the paper does not analyse the influence of performance if various public datasets are used.

---

> ### Author Rebuttal · Authors · 2026-03-31
>
> Thank you for your positive assessment and helpful suggestions.
>
> **Weakness 1 (Language-Only Evaluation).**
>
> Our focus on NLP tasks is motivated by the most critical deployment scenarios for privacy-sensitive fine-tuning. The framework itself is not linguistically constrained, as detailed in Q1 below.
>
> **Q1 (Multimodal Applicability).**
>
> PISA's three mechanisms generalize to other modalities. MRP enforces manifold alignment via a reconstruction objective, directly replaceable by MAE loss for vision inputs. DSC constructs a residual patch from the client's clean local input to correct noisy server features; for vision inputs, token embeddings are replaced by patch embeddings (e.g., ViT patch projections). UGR requires a confidence estimator suited to the modality; for vision-language models such as LLaVA, cross-modal alignment scores (e.g., CLIP similarity between image and text representations) provide a natural substitute for the MLM-based posterior. The primary open challenge is defining a unified LDP mechanism across modalities with heterogeneous input spaces, which we plan to address in future work.
>
> **Q2 (Different Model Structures).**
>
> The LLaMA-2-7B results below directly address this question. PISA achieves consistent gains over Hosted-LDP on both encoder-only (BERT) and decoder-only (LLaMA-2-7B) architectures, confirming that MRP, DSC, and UGR are not tied to any specific model structure.
>
> **Weakness 2 (Single Model).**
>
> We extend experiments to LLaMA-2-7B (split = 24 layers Head / 8 layers Tail; metric = Accuracy (%)):
>
> | Method (SST-2)      | ε=12 | ε=16 |
> | ------------------- | ---- | ---- |
> | Vanilla FT (Non-DP) | 93.5 | 93.5 |
> | Naive-RS            | 50.4 | 50.4 |
> | Hosted-LDP          | 68.7 | 90.6 |
> | PISA (Ours)         | 75.6 | 86.2 |
>
> | Method (QNLI)       | ε=12 | ε=16 |
> | ------------------- | ---- | ---- |
> | Vanilla FT (Non-DP) | 92.5 | 92.5 |
> | Naive-RS            | 51.1 | 51.1 |
> | Hosted-LDP          | 62.2 | 88.7 |
> | PISA (Ours)         | 65.0 | 75.2 |
>
> Under strict privacy (ε=12), PISA consistently outperforms Hosted-LDP across both datasets and architectures. The crossover at ε=16 is consistent with Section 5.2 and expected under our dual-constraint design.
>
> **Weakness 3 & Q3 (Public Dataset Effect).**
>
> Among our baselines, Vanilla FT and Naive-RS do not apply LDP and therefore have no privacy-induced distribution shift to correct, making MRP inapplicable to both. Hosted-LDP applies LDP with full-parameter server-side fine-tuning, making it the only meaningful baseline for this comparison. To isolate the effect of public data access, we equip Hosted-LDP with the same MRP pre-training on Wikipedia (BERT-base, ε=12, SST-2):
>
> | Method         | Public Pre-training | ε=12 | ε=14 |
> | -------------- | ------------------- | ---- | ---- |
> | Hosted-LDP     | No                  | 59.4 | 86.4 |
> | Hosted-LDP+MRP | Yes                 | 75.4 | 85.2 |
> | PISA           | Yes                 | 84.5 | 87.1 |
>
> MRP alone brings Hosted-LDP from 59.4 to 75.4 (+16.0) at ε=12, confirming that manifold alignment is independently effective. At ε=14, Hosted-LDP+MRP slightly underperforms Hosted-LDP, as MRP's denoising objective adds marginal constraint when noise is low. PISA consistently outperforms both and, critically, neither Hosted-LDP variant satisfies our model IP constraint. We will include Hosted-LDP+MRP as an additional baseline in the camera-ready version.
>
> By design, MRP's core objective is minimizing $\mathcal{L}\_{\text{Denoise}}(h\_{\text{pred}}, h\_{\text{clean}})$, which optimizes the geometric relationship between perturbed and clean representations rather than task-specific semantics. This means MRP learns a general manifold alignment capability that is not tied to the domain of the public dataset, and any sufficiently diverse natural language corpus provides adequate coverage. We verify this empirically with two public datasets of contrasting relevance (ε=12): Wikipedia (broad encyclopedic text) and OpenWebText (web-crawled text with higher relevance to sentiment and NLI tasks):
>
> | Public Data        | SST-2 | QNLI | MNLI |
> | ------------------ | ----- | ---- | ---- |
> | Wikipedia Corpus   | 84.5  | 68.2 | 69.8 |
> | OpenWebText Corpus | 82.5  | 69.4 | 71.2 |
>
> Performance varies by at most 2.0 points across all tasks, confirming that PISA is robust to public dataset choice.
>
> At last, we sincerely appreciate your valuable feedback, and we will carefully consider all your suggestions to further improve our paper. Thank you very much!

---

> > ### Author Rebuttal · Reviewer_2NiV · 2026-04-02
> >
> > Thank you for the responses. The rebuttal has addressed my main concerns.
> >
> > Therefore, I have adjusted my overall recommendation accordingly.

---

> > > ### Author Response · Authors · 2026-04-02
> > >
> > > Thank you for your positive feedback and for adjusting your score. We are glad the clarifications were helpful and will carefully incorporate all your suggestions in the revision.

---

### Official Review · Reviewer_C5Na · 2026-03-09

**Soundness:** 3
**Presentation:** 3
**Significance:** 3
**Originality:** 3
**Overall Recommendation:** 4
**Confidence:** 3

**Summary:**

The paper presents PISA, a split-learning framework designed to fine-tune Large Language Models while simultaneously preserving data privacy via LDP and protecting model intellectual property via a gradient blockage protocol. The framework introduces three main components: Manifold rectification pre-training to build server-side robustness against noise, dual-stream semantic compensation to inject clean local features, and Utility-aware gradient rectification to re-weight gradients based on server-computed confidence scores. The authors evaluate their framework on GLUE benchmarks and demonstrate performance improvements over a standard privacy-prioritized baseline under strict privacy budgets.

**Compliance With Llm Reviewing Policy:**

Affirmed.

**Final Justification:**

The authors addressed my concerns and I have increased my score.

**Key Questions For Authors:**

- How exactly are MRP and UGR (specifically the confidence score $\alpha$ in Equation 7) implemented for Llama-2-7B, given that causal LLMs do not possess an MLM head?
- What is the downstream task utility of Llama-2-7B under the PISA framework? Does the frozen-head architecture allow a 7B model to converge meaningfully on standard benchmarks?
- How does PISA compare against parameter-efficient fine-tuning with DP-SGD in terms of both privacy-utility trade-off and computational efficiency?
- How sensitive is the Manifold Rectification Pre-training to domain shifts between the server's public data and the client's private downstream data?

**Limitations:**

Yes.

**Strengths And Weaknesses:**

Strengths

- The strict gradient blockage combined with forward-pass LDP provides a robust structural defense against both data reconstruction and model IP theft.
- The efficiency analysis in Table 5 demonstrates that the framework effectively reduces the memory footprint, enabling the processing of larger models on consumer-grade GPUs, which is a highly practical contribution.

Weaknesses
- The paper frames itself as a solution for "Large Language Models". However, all utility experiments exclusively use BERT-base. Demonstrating utility on a 110M parameter encoder model from 2018 is insufficient to support claims about modern LLMs.
- The Manifold Rectification Pre-training and Utility-Aware Gradient Rectification explicitly depend on Masked Language Modeling. Equation 7 uses $\mathcal{H}_{MLM}$ to compute confidence scores. It is mathematically ambiguous how this framework is adapted for modern decoder-only causal language models.
- While the authors report system efficiency for Llama-2-7B in Table 5, they provide zero evidence that the model actually converges or achieves reasonable task utility under the PISA framework. An efficiency speedup is irrelevant if the model's accuracy collapses.
- Evaluating an LLM framework solely on classification tasks is a severe limitation. It completely ignores generative capabilities, which represent the primary function of modern language models.

---

> ### Author Rebuttal · Authors · 2026-03-31
>
> Thank you for your review. We clarify the architectural generality of PISA and provide additional validation on modern causal LLMs below.
>
> **Weakness 2 & Q1 (MLM Dependency).**
>
> PISA's design extends naturally to decoder-only architectures. MRP and UGR do not rely on MLM as a mechanism. Both rely on the server head's ability to model the natural language distribution, a property shared by any competent LM. The adaptation to causal architectures follows directly from the framework's design, as we clarify below.
>
> For MRP, $\mathcal{L}\_{\text{MLM}}$ is a specific instantiation of a general reconstruction objective. For causal LMs, it is naturally replaced by $\mathcal{L}\_{\text{CLM}}(\tilde{x}) = -\frac{1}{L}\sum\_{t=1}^{L} \log P(x\_t \mid x\_{<t})$, while $\mathcal{L}_{\text{Denoise}}$ remains unchanged as it imposes no assumption on the LM objective type. The Phase I objective preserves the same structural form as Eq. 3:
>
> $$\mathcal{L}\_{\text{Phase I}} = \mathcal{L}\_{\text{CLM}}(\tilde{x}) + \lambda \mathcal{L}\_{\text{Denoise}}(h\_{\text{pred}}, h\_{\text{clean}})$$
>
> For UGR, $H\_{\text{MLM}}$ in Eq. 7 is a confidence estimator that assesses semantic integrity of noisy inputs. For causal LMs, the LM head trained via $\mathcal{L}\_{\text{CLM}}$ during MRP serves the same role and is retained on the server after the split. At fine-tuning time, $h\_{\text{noisy}}$ (the output of the $K$-th Head layer, e.g., $K{=}24$ for LLaMA-2-7B) is passed through the remaining $L{-}K$ frozen layers and the LM head in a no-gradient forward pass, producing a sequence-level confidence score:
>
> $$\alpha = \exp\\!\left(\frac{1}{L}\sum_{t=1}^{L} \log P(x_t \mid x_{<t})\right) \in (0,1]$$
>
> This generalizes Eq. 7 from single-token max posterior to sequence-level average log-likelihood, preserving the same range $(0,1]$ and monotonicity. The no-gradient forward pass introduces negligible overhead.
>
> **Weakness 1, 3 & Q1, Q2 (Architectural Compatibility and Utility on LLaMA-2-7B).**
>
> To empirically confirm the above, we evaluated PISA on LLaMA-2-7B under strict privacy constraints. The model converges stably across all tested budgets. Conditions: backbone = LLaMA-2-7B; split = 24 layers Head / 8 layers Tail; metric = Accuracy (%); averaged over 3 runs.
>
> | Method (SST-2)      | ε=12 | ε=16 |
> | ------------------- | ---- | ---- |
> | Vanilla FT (Non-DP) | 93.5 | 93.5 |
> | Naive-RS            | 50.4 | 50.4 |
> | Hosted-LDP          | 68.7 | 90.6 |
> | PISA (Ours)         | 75.6 | 86.2 |
>
> | Method (QNLI)       | ε=12 | ε=16 |
> | ------------------- | ---- | ---- |
> | Vanilla FT (Non-DP) | 92.5 | 92.5 |
> | Naive-RS            | 51.1 | 51.1 |
> | Hosted-LDP          | 62.2 | 88.7 |
> | PISA (Ours)         | 65.0 | 75.2 |
>
> Under strict privacy (ε=12), PISA consistently outperforms Hosted-LDP on both datasets, while Naive-RS fails to converge. The crossover at ε=16 mirrors Figure 2 and is expected, as relaxed noise allows Hosted-LDP's full-parameter advantage to emerge at the cost of model IP exposure.
>
> **Weakness 4 (Generative Evaluation).**
>
> We clarify that the empirical claims of this work are scoped to privacy-preserving collaborative adaptation under discriminative downstream fine-tuning, and we do not claim full generative LLM coverage in this work. The LLaMA-2-7B results above confirm that the framework extends to decoder-only architectures under strict privacy constraints. While PISA is compatible with generative tasks, achieving strong generative performance under strict LDP constraints remains challenging and warrants dedicated evaluation and mechanism refinement, which we identify as an important direction for future work.
>
> **Q3 (vs. DP-SGD + PEFT).**
>
> DP-SGD and PISA target mutually exclusive threat models: DP-SGD relies on a trusted central server, which is a strong assumption absent in our LDP threat model where the server is explicitly untrusted. Furthermore, DP-SGD + PEFT requires adapted parameters to reside on the server, directly violating our model IP constraint. For reference, DP-SGD + LoRA (r=8) on LLaMA-2-7B at ε=6 achieves 90.6% on SST-2 and 84.8% on QNLI, but incurs 30,421 MB client VRAM and exposes model IP, both of which are unacceptable under our threat model.
>
> **Q4 (Public-Data Sensitivity).**
>
> We add ablations with different public datasets (ε=12, BERT-base):
>
> | Public Data        | SST-2 | QNLI | MNLI |
> | ------------------ | ----- | ---- | ---- |
> | Wikipedia Corpus   | 84.5  | 68.2 | 69.8 |
> | OpenWebText Corpus | 82.5  | 69.4 | 71.2 |
>
> These results suggest preliminary robustness to public dataset choice.
>
> At last, we sincerely appreciate your valuable feedback, and we will carefully consider all your suggestions to further improve our paper. We hope these clarifications and additional results have addressed your primary concerns. Thank you very much!

---

> > ### Author Rebuttal · Reviewer_C5Na · 2026-04-01
> >
> > I appreciate the authors' efforts to address the concerns raised, but I still have the following concerns:
> >
> > - The transition from token-level confidence in BERT to sequence-level average log-likelihood for Causal LMs is a significant conceptual shift. The authors provide no theoretical or empirical evidence that a sequence-level scalar can effectively substitute for token-level gating in identifying local semantic distortions caused by LDP noise. This lack of rigorous equivalence undermines the robustness of the UGR mechanism across different architectures.
> >
> > - The response to Reviewer rQ4D reveals that the reported performance gains are heavily reliant on the MRP pre-training with public data. When baselines are granted the same data access, the unique architectural advantage of PISA appears marginal.
> >
> > - I still see a mismatch between the paper’s framing and its evidence. The submission is presented broadly as an LLM fine-tuning framework, but the main paper’s experiments remain centered on BERT-base and GLUE-style classification. The rebuttal adds only two LLaMA-2-7B classification results, and the authors now explicitly state that their empirical claims are limited to discriminative downstream fine-tuning rather than generative evaluation. This still feels narrower than the overall framing of the paper.
> >
> > - The decoder-only implementation of UGR remains somewhat unclear to me. In the paper, the upper layers are detached and sent to the client as the Tail, while the server retains only the frozen Head. But in the rebuttal, the confidence score for causal LMs is computed by passing the Head output through the remaining frozen layers and the LM head on the server. I would appreciate a more precise explanation of where these layers reside after the split, and whether this changes the deployment boundary or efficiency assumptions
> >
> > I would consider to raise my score if these issues can be further clarified.

---

> > > ### Author Response · Authors · 2026-04-02
> > >
> > > Thank you for your review. We clarify each concern below.
> > >
> > > **Concern 1 (Sequence-level vs. Token-level Confidence for Causal LMs).**
> > >
> > > The transition to sequence-level confidence is an architecture-aware design, not an ad hoc substitution. In bidirectional encoder models such as BERT, attention allows each position to observe full context, making token-level max posterior sensitive to local distortions. In causal LMs, each position only attends to left-side context, making token-level max posterior blind to distortions in later positions. This limitation is more severe under strict privacy budgets where more tokens are perturbed. Sequence-level average log-likelihood aggregates global semantic information across the full sequence, directly compensating for this structural limitation.
> > >
> > > We provide empirical evidence on LLaMA-2-7B:
> > >
> > > | Method (SST-2)             | ε=12 | ε=16 |
> > > | -------------------------- | ---- | ---- |
> > > | UGR (token-level)          | 72.2 | 85.5 |
> > > | UGR (sequence-level, Ours) | 75.6 | 86.2 |
> > >
> > > | Method (QNLI)              | ε=12 | ε=16 |
> > > | -------------------------- | ---- | ---- |
> > > | UGR (token-level)          | 61.7 | 73.9 |
> > > | UGR (sequence-level, Ours) | 65.0 | 75.2 |
> > >
> > > The gap is larger under strict privacy (ε=12: +3.4/+3.3 points) than relaxed privacy (ε=16: +0.7/+1.3 points), consistent with our analysis: heavier perturbation amplifies the token-level estimator's structural blind spot, while sequence-level aggregation remains robust. This confirms that sequence-level confidence is not merely equivalent to token-level confidence, but strictly more appropriate for causal architectures under LDP noise.
> > >
> > > **Concern 2 (Architectural Advantage and MRP Contribution).**
> > >
> > > This comparison conflates two fundamentally different constraint settings. Hosted-LDP and Hosted-LDP+MRP both require fine-tuned parameters to reside on the server, directly violating our model IP constraint. They represent an upper-bound reference achievable only by abandoning IP protection, not architectural competitors to PISA.
> > >
> > > We acknowledge that MRP independently contributes to utility under LDP, as evidenced by Hosted-LDP+MRP improving from 59.4% to 75.4% at ε=12. However, even with the same public data access, PISA still outperforms Hosted-LDP+MRP by 9.1 points at ε=12, demonstrating that the architectural design contributes beyond MRP alone. More critically, neither Hosted-LDP nor Hosted-LDP+MRP satisfies our model IP constraint, as both expose fine-tuned parameters to the server. They address only data privacy, not the dual-constraint problem PISA targets.
> > >
> > > The meaningful comparison is within the dual-constraint setting. We additionally tested Naive-RS under LDP noise, which hovers around 50% on SST-2 at ε=10, effectively random guessing. In contrast, PISA achieves 81.2%, a gain of over 30 points. We note that equipping Naive-RS with both LDP noise and MRP pre-training on public data corresponds to our ablation variant PISA- in Table 1, which still significantly underperforms full PISA, confirming that the performance gains stem from the complete framework rather than public data access alone.
> > >
> > > **Concern 3 (Framing vs. Evidence Mismatch).**
> > >
> > > We agree with this concern. We will revise the framing to more explicitly reflect the current empirical scope. We also note that extending PISA to generative evaluation presents a non-trivial technical challenge: DSC's semantic compensation is applied once during prefill and cannot propagate corrections to newly generated tokens during autoregressive decoding. This requires designing a dedicated decoding-aware compensation mechanism, which we identify as an important direction for future work. We thank the reviewer for highlighting this priority.
> > >
> > > **Concern 4 (Deployment Boundary of Decoder-only UGR).**
> > >
> > > After Phase I, the model is split at K=24 for LLaMA-2-7B. The server retains the bottom 24 layers as the frozen Head. The upper 8 layers and classification head are transmitted to the client as the trainable Tail. To enable UGR, the server additionally retains a separate frozen copy of the upper 8 layers and the LM head from MRP, used exclusively for computing α with no gradient updates, imposing no additional IP risk.
> > >
> > > During fine-tuning, the server computes hnoisy via the Head, passes it through this frozen copy in a no-gradient forward pass to obtain α, and transmits both to the client in a single communication round with no additional synchronization overhead. The extra inference over 8 frozen layers is negligible given server-side resources, as it involves only a no-gradient forward pass over a small frozen module. Task-specific trainable parameters remain exclusively on the client, and the frozen server copy serves only as a read-only confidence estimator, leaving the deployment boundary unchanged.

---

### Official Review · Reviewer_reqN · 2026-03-13

**Soundness:** 3
**Presentation:** 4
**Significance:** 3
**Originality:** 3
**Overall Recommendation:** 4
**Confidence:** 3

**Summary:**

This paper studies privacy-preserving collaborative fine-tuning of large language models under a setting where both data privacy and model intellectual property (IP) must be protected. The authors consider a reverse split learning architecture where the server hosts the backbone model and the data holder maintains the task-specific Tail, while gradients are blocked to prevent IP leakage. To address the resulting utility degradation in this setting, the paper proposes PISA, a structured framework for privacy-preserving split learning. Experiments on several GLUE benchmarks show that the proposed approach improves performance under strict privacy budgets while maintaining the structural protection of both data privacy and model IP.

**Compliance With Llm Reviewing Policy:**

Affirmed.

**Final Justification:**

I appreciate the rebuttal for addressing my concerns, including support for decoder-only models and convergence evidence. While some limitations remain, they do not affect my overall view. I keep my weak accept rating.

**Key Questions For Authors:**

Q1. How would the proposed framework apply to decoder-only architectures, which are more common in modern LLM systems?

Q2. Could you provide empirical evidence (e.g., training curves or gradient statistics) to validate whether the predicted convergence behavior is observed in practice?

Q3. Have the authors explored alternative confidence estimators or scoring functions of UGR, and how sensitive is the method to this design choice?

**Limitations:**

- The DSC mechanism relies on constructing representations from the clean local input, which requires the client to maintain part of the embedding or representation pipeline. In practice, this assumption may limit the applicability of the framework, since some deployment settings may not allow the client to access or maintain such components of the model.

- The privacy guarantee of the framework largely relies on the underlying LDP mechanism rather than the proposed architecture itself. The paper would benefit from a clearer discussion of whether the proposed framework introduces any additional privacy guarantees beyond those provided by the sanitization mechanism.

**Strengths And Weaknesses:**

Strengths

- The paper addresses the problem of jointly protecting data privacy and model IP during collaborative fine-tuning, which is an important and practically relevant setting.

- This paper clearly identifies the “Frozen-Head, Limited-Tail” bottleneck caused by gradient blockage in reverse split learning. The proposed components target different sources of utility degradation under this constraint.

- The experimental results show that the proposed method improves performance in strict privacy regimes where LDP noise significantly degrades model utility.

Weaknesses

- All experiments are conducted using BERT-base as the backbone. It is unclear whether the proposed framework scales well to larger models or different architectures. For example, many modern LLM systems are decoder-only models.

- The paper provides a theoretical convergence analysis for the UGR mechanism. However, the paper does not provide empirical evidence validating the convergence behavior predicted by the theory.

---

> ### Author Rebuttal · Authors · 2026-03-31
>
> Thank you for recognizing the importance of the problem and for your constructive suggestions.
>
> **Weakness 1 & Q1 (Architectural Compatibility to Decoder-Only Architectures).**
>
> PISA's design extends naturally to decoder-only architectures. MRP and UGR do not rely on MLM as a mechanism. Both rely on the server head's ability to model the natural language distribution, a property shared by any competent LM.
>
> For MRP, $\mathcal{L}\_{\text{MLM}}$ is replaced by $\mathcal{L}\_{\text{CLM}}(\tilde{x}) = -\frac{1}{L}\sum\_{t=1}^{L}\log P(x\_t \mid x\_{<t})$, while $\mathcal{L}\_{\text{Denoise}}$ remains unchanged. The Phase I objective preserves the same structural form as Eq. 3:
>
> $$\mathcal{L}\_{\text{Phase I}} = \mathcal{L}\_{\text{CLM}}(\tilde{x}) + \lambda \mathcal{L}_{\text{Denoise}}(h\_{\text{pred}}, h\_{\text{clean}})$$
>
> For UGR, the LM head trained via $\mathcal{L}\_{\text{CLM}}$ during MRP is retained on the server and replaces $H\_{\text{MLM}}$. At fine-tuning time, $h\_{\text{noisy}}$ is passed through the remaining $L{-}K$ frozen layers and the LM head in a no-gradient forward pass, producing:
>
> $$\alpha = \exp\\!\left(\frac{1}{L}\sum\_{t=1}^{L} \log P(x\_t \mid x\_{<t})\right) \in (0,1]$$
>
> This preserves the same range and monotonicity as Eq. 7 with negligible overhead. To empirically confirm this, we evaluate on LLaMA-2-7B (split = 24 layers Head / 8 layers Tail; metric = Accuracy (%); averaged over 3 runs):
>
> | Method (SST-2)      | ε=12 | ε=16 |
> | ------------------- | ---- | ---- |
> | Vanilla FT (Non-DP) | 93.5 | 93.5 |
> | Naive-RS            | 50.4 | 50.4 |
> | Hosted-LDP          | 68.7 | 90.6 |
> | PISA (Ours)         | 75.6 | 86.2 |
>
> | Method (QNLI)       | ε=12 | ε=16 |
> | ------------------- | ---- | ---- |
> | Vanilla FT (Non-DP) | 92.5 | 92.5 |
> | Naive-RS            | 51.1 | 51.1 |
> | Hosted-LDP          | 62.2 | 88.7 |
> | PISA (Ours)         | 65.0 | 75.2 |
>
> Under strict privacy (ε=12), PISA consistently outperforms Hosted-LDP on both datasets, while Naive-RS fails to converge. The crossover at ε=16 mirrors Figure 2 and is expected, as relaxed noise allows Hosted-LDP's full-parameter advantage to emerge at the cost of model IP exposure.
>
> **Weakness 2 & Q2 (Empirical Convergence Validation).**
>
> We provide training curves and gradient variance statistics to empirically validate Theorem 4.2 (SST-2, ε=12, Split@Layer9, BERT-base). As shown in the linked figure, UGR reduces gradient norm variance by 46% (σ: 0.84 → 0.45) and achieves faster early convergence (85.6% vs 82.1%), yielding a smoother optimization trajectory consistent with our $\mathcal{O}(1/\sqrt{T})$ bound.
>
> [Anonymous link to convergence plots: [/r/ICML2026-rebuttal-4EEE/](https://anonymous.4open.science/r/ICML2026-rebuttal-4EEE/)]
>
> **Q3 (Sensitivity of UGR Confidence Estimators).**
>
> We tested alternative confidence estimators (BERT-base, ε=12):
>
> | Confidence Metric              | SST-2 | QNLI | Optimization Stability |
> | ------------------------------ | ----- | ---- | ---------------------- |
> | Max Posterior (Eq. 7, Default) | 84.5  | 68.2 | High                   |
> | Entropy-based                  | 82.2  | 62.1 | Medium                 |
>
> The performance gap is modest (≤2.3 points on SST-2, ≤6.1 on QNLI), indicating that UGR remains effective under alternative confidence estimators in our tested settings. Max posterior probability performs best as it most directly reflects the semantic integrity of the noisy input under the server's linguistic prior.
>
> At last, we sincerely appreciate your valuable feedback, and we will carefully consider all your suggestions to further improve our paper. Thank you very much!

---

> > ### Author Rebuttal · Reviewer_reqN · 2026-04-02
> >
> > Thanks for your rebuttal. I will maintain my score.

---

> > > ### Author Response · Authors · 2026-04-02
> > >
> > > Thank you for your feedback. We are glad the clarifications were helpful and will carefully incorporate all your suggestions in the revision.

---

### Decision · Program_Chairs · 2026-04-30

**Decision:**

Accept (regular)

**Comment:**

This paper proposes PISA, a split fine-tuning framework for LLMs that aims to jointly preserve data privacy and model intellectual property through a combination of manifold rectification, dual-stream compensation, and utility-aware gradient rectification.

During the rebuttal, the authors addressed key concerns by providing additional experiments on decoder-only models, empirical validation of convergence behavior, and ablations on the use of public data. Following these clarifications, reviewers indicated that their concerns were largely resolved and maintained or raised their scores to weak accept or accept.

Remaining concerns include the lack of evaluation on generative tasks. While this can be addressed in future work, I suggest that the authors revise the framing of the paper and clarify that the current scope focuses on classification tasks, as discussed.

Overall, based on the reviewers’ assessments and updates after rebuttal, I recommend acceptance.